# CrossMAE: Decoupled Masked Image Modeling with Cross Prediction

## Abstract

We present CrossMAE, a novel and flexible masked image modeling (MIM) approach, which decouples the existing *mask-then-predict* objective in MIM progressively into *mask-then-draw* and *draw-then-predict*. During the *mask-then-draw* phase, an auxiliary drawing head models the uncertainty and produces coarse and diverse outputs. Subsequently in *draw-then-predict*, the backbone receives the completions and strives to predict versatile signals from them. These two decoupled objectives are end-to-end trainable and involved in a single pass, splitting low-level generation and high-level understanding. Through extensive experiments and compelling results on a variety of tasks, we demonstrate that the proposed pre-training scheme learns generalizable features effectively, including image classification, semantic segmentation, object detection, instance segmentation, and even facial landmark detection. Beyond surpassing existing MIM counterparts, CrossMAE exhibits better data efficiency, in both pre-training and fine-tuning.

## 1 Introduction

With the tremendous success in natural language processing, the transformer architecture has gradually become a prevalent option in computer vision. ViT (Dosovitskiy et al., 2020) mimics the sequence modeling design and shows great scalability to large-scale training data, leading to a phenomenal success. However, due to low locality inductive bias (d'Ascoli et al., 2021; Lee et al., 2021), vision transformers often yield lowered performance on mid-sized datasets (e.g. ImageNet (Deng et al., 2009)). Therefore, self-supervised pre-training schemes, such as MoCov3 (Chen et al., 2021) and DINO (Caron et al., 2021), are proposed to unleash the capability of vision transformers. Among them, masked image modeling (MIM)-based methods (He et al., 2021; Bao et al., 2021; Xie et al., 2022; Chen et al., 2022; El-Nouby et al., 2021; Dong et al., 2021) exhibit potentials on learning transferable features for downstream tasks.

Originated from masked language modeling in BERT (Devlin et al., 2019), MIM follows the conceptually simple *mask-then-predict* idea, masking a portion of input and then predicting the masked content. However, the inherent difference between image and text requires domain knowledge to adapt the underlying idea. To imitate the pre-existing vocabulary and discrete tokens in MLM, discrete VAEs (Ramesh et al., 2021) are leveraged in BERT pre-training of image transformers (Bao et al., 2021) to process continuous pixels into discrete codes, requiring the target model to predict tokenized features. Masked Autoencoders (He et al., 2021) and SimMIM (Xie et al., 2022) further demonstrate the viability of reconstructing the missing pixels. In an effort to find valid objectives beyond raw pixels, recent works alternatively exploit different feature extractors as guidance, such as hand-crafted HOG descriptor (Wei et al., 2021), online network (Zhou et al., 2021), or pre-trained models (Wei et al., 2021; Dong et al., 2021).

Despite the upsurge of this *mask-and-predict* pre-training scheme, the debate about finding a proper objective in MIM as well as its comparison with foregoing methods such as contrastive learning still goes on. With two questions remains to be explored: *what does MIM learn*, and *how does it benefit downstream tasks*, we strive to dissect MIM methods from these two perspectives:

1. Humans have an uncanny ability to sense the rich visual world, even with partially visible content. Similarly, the idea behind denoising autoencoder (Vincent et al., 2008) and masked signal modeling

is to obtain better representation through such capability. However, when given an image with a high masking ratio (e.g. 75%), human can easily imagine different outputs. Existing MIM frameworks only consider one-to-one mapping and discard the diversity, which is sub-optimal for learning generalizable feature and possibly leads to "memorization" (Feng et al., 2021a; Webster et al., 2019).

2. The generative supervisory signals bring inconsistency to pre-training and downstream tasks, resulting in a significantly low performance without fine-tuning. For instance, while MAE (He et al., 2021) reports state-of-the-art image classification accuracy after the backbone being fine-tuned, exploiting frozen features from its pre-trained model performs significantly worse than previous contrastive or self-distillation counterparts such as DINO (Caron et al., 2021). As contrastive method has been analyzed to bring alignment and uniformity on features (Wang & Isola, 2020), MIM pre-training merely focuses on reconstruction and poses no such explicit constraints. Meanwhile, many works (Desai & Johnson, 2021; Wei et al., 2021) conjecture that the noisy and redundant pixel supervision affects the tendency to learn rich semantic features. Moreover, masked inputs incur a discrepancy between pre-training and fine-tuning, as the fine-tuning tasks rarely see mask tokens (Bao et al., 2021) or incomplete image patches (He et al., 2021). Therefore, it's crucial to align closer between the objectives of MIM and downstream understanding.

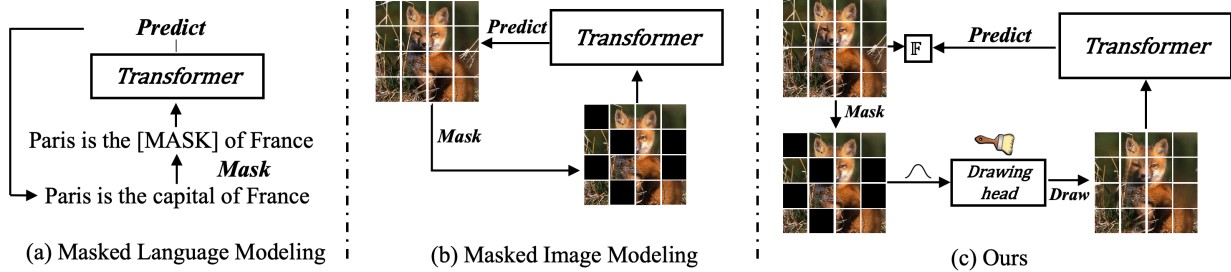

Figure 1: Illustration of the differences among MLM, MIM, and our proposed CrossMAE. The predictor, decoder and tokenization are omitted for simplicity. In contrast to existing MLM and MIM methods that generally follow *mask-then-predict*, CrossMAE decouples the overall target into the sequential *mask-then-draw* and *draw-then-predict*. During the drawing step, the noise denotes modeling the one-to-many mapping given masked image. $\mathbb{F}$ refers to versatile targets for the backbone to predict.

In this work, we present CrossMAE, a pre-training objective that decouples existing masked image modeling into *mask-then-draw* and *draw-then-predict* steps. Rather than directly pushing the backbone to reconstruct missing content as in MIM, CrossMAE prepends an auxiliary drawing head with noise injected, making the original backbone focus on "understand" the coarsely-completed outputs through cross prediction. Specifically, we do not reckon the drawing module to reconstruct signals accurately. In contrast, it intends to model the uncertainty and is supervised with smoothed weak signals. Whereas the completions contain imperfect texture and distortions, the backbone plays a different role in understanding their potential contexts. CrossMAE not only mitigates the pre-training-fine-tuning discrepancy, but also strengthens the backbone with invariance towards changing shape-texture, which is commonly desired in vision architectures (Touvron et al., 2022; Li et al., 2021b).

To conclude, CrossMAE offers a novel perspective for masked image modeling by decoupling generation and understanding as shown in Figure 1. With extensive experiments on image classification, object detection, instance segmentation, semantic segmentation, and facial landmark detection, it demonstrates superiority over existing MIM frameworks, either being fine-tuned or serving as a frozen feature extractor. Compared with *mask-then-predict* baselines like MAE (He et al., 2021), the results of fine-tuning and linear probing on ImageNet-1k are improved by 1% and 8.5%, with 300 pre-training epochs. Our proposed CrossMAE even adapts seamlessly to facial landmark detection, reducing the failure rate by half. Beyond generalization, we quantitatively show that CrossMAE benefits data efficiency, in both pre-training and transfer learning.

## 2    Related Works

**Vision Transformer**    Firstly caught attention in machine translation (Vaswani et al., 2017), transformers have gradually became a primary architecture for generic language understanding. DETR (Carion et al., 2020) and ViT (Dosovitskiy et al., 2020) later demonstrate the potentials of adopting a pure transformer in detection and image classification. In order to generalize better for vision tasks, data augmentations (Touvron et al., 2021; 2022), hybrid designs (Xu et al., 2021; Wang et al., 2021a; Pan et al., 2021; Wu et al., 2021), and convolutional stems (Xiao et al., 2021; Wang et al., 2021b) are incorporated into the naive architecture. Local-global attention (Liu et al., 2021b; Vaswani et al., 2021; Yang et al., 2021; Han et al., 2021) is studied to reduce the computation cost. Recently, vision transformers have been investigated with their performance on downstream tasks, such as image restoration (Liang et al., 2021; Esser et al., 2021), detection (Li et al., 2022; Zhu et al., 2021), semantic segmentation (Zheng et al., 2021; Bao et al., 2021), and video recognition (Arnab et al., 2021; Bertasius et al., 2021).

**Self-supervised Pre-training**    Pre-training plays a critical role in existing transformer-based frameworks, where self-supervised methods begin to shine. Contrastive and generative (i.e. masked signal modeling) pre-training serve as two major directions. As SimCLR (Chen et al., 2020b) and MoCo (He et al., 2020) adopt instance discrimination (Wu et al., 2018c) to learn the invariance between augmented views and obtain superior performance, MoCov3 (Chen et al., 2021) and DINO (Caron et al., 2021) explore to train ViT with contrastive learning or self-distillation. While contrastive and distillation method constrains the feature space and brings nice linear separability (Wang & Isola, 2020), the other direction, masked language modeling, has witnessed great success in NLP (Devlin et al., 2019; Radford et al., 2018; Lan et al., 2019). The idea of masked image modeling was introduced to vision tasks in BEiT (Bao et al., 2021). MAE (He et al., 2021) and SimMIM (Xie et al., 2022) further investigated encoder-decoder design and showed the potentials of pixel reconstruction in representation learning. As PeCo (Dong et al., 2021) and MaskFeat (Wei et al., 2021) improved MIM either from better tokenizer or reconstruction targets, iBOT (Zhou et al., 2021), data2vec (Baevski et al., 2022) and SplitMask (El-Nouby et al., 2021) also exploited useful strategies from contrastive methods. Unlike existing frameworks that train the backbone to generate missing information, our work disentangles MIM into *mask-then-draw* and *draw-then-predict* stages where the backbone focuses more on understanding the contents and involves computing the visual invariance. Note that our method is versatile with different MIM frameworks or objectives.

**Generative Modeling**    Our method also connects to generative modeling. According to (Vincent et al., 2008), masked autoencoders can be illustrated from a generative model perspective, capturing the unknown distributions of its observed inputs. Given a largely-masked image, there certainly exist multiple targets that are semantically meaningful. However, MIM is currently treated deterministically, as either a regression (He et al., 2021; Xie et al., 2022) or classification (Bao et al., 2021; Dong et al., 2021) question given target inputs. Similar studies have been investigated in preventing the GAN replication (or memorization (Feng et al., 2021a; Webster et al., 2019)) problem. To generate diverse outputs, there exists many options such as autoregressive (Radford et al., 2018; Chen et al., 2020a), VAE (Kingma & Welling, 2013), GANs (Goodfellow et al., 2014; Brock et al., 2018). Instead of generating pluralistic results as in diverse image inpainting (Liu et al., 2021a), our goal is to benefit from varying input contents during pre-training. Motivated by noise injection (Feng et al., 2021b; Karras et al., 2019) in GANs, we feed noise vectors into the auxiliary drawing head. Having limited capacity and supervised with smoothed signals, the module produces coarse and diverse outputs. During pre-training, the incorporated stochasticity enriches the feature learning of the backbone.

## 3    Proposed Framework: CrossMAE

The proposed framework consists of two major modules. One refers to the prepended drawing head, while the other can be the backbone transformer to be trained. As illustrated in Figure 2, we disentangle the pipeline of masked image modeling into generation and understanding. Note that the auxiliary head is lightweight compared to the backbone, and the whole system is end-to-end trainable. During the first phase, we focus on *mask-then-draw* and model the one-to-many mapping, giving coarse completions to masked images. In

the second phase of *mask-then-predict*, diverse and ambiguous images are available for boosting both the semantic understanding and visual invariance of the backbone.

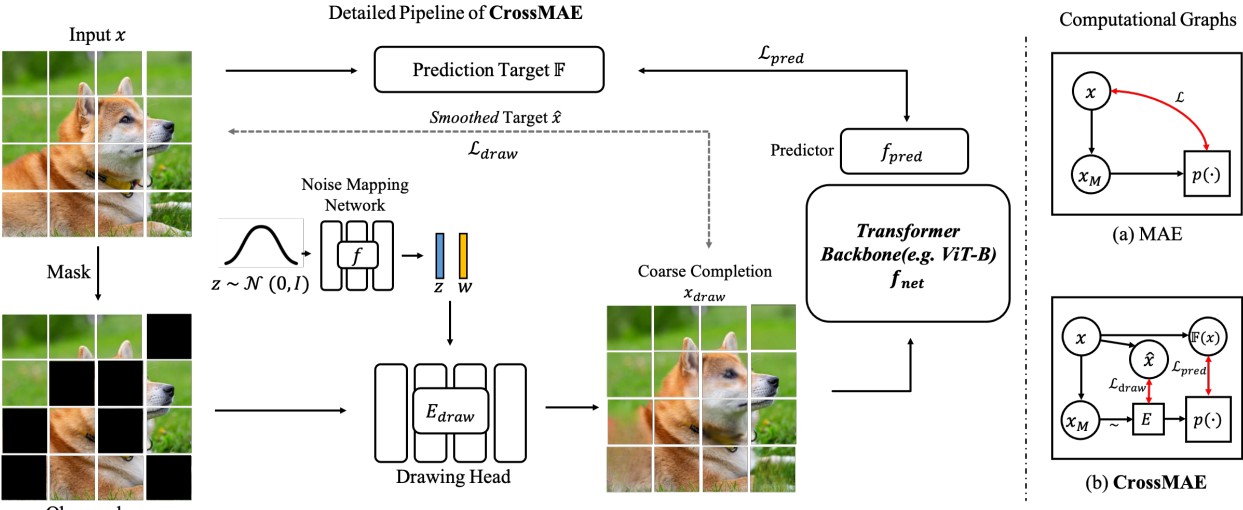

Figure 2: Our framework. The left portion demonstrates the detailed pipeline of CrossMAE. On the right, we provide two simplified computational graphs of MAE (He et al., 2021) and our method to illustrate the difference. For simplicity, $p(\cdot)$ includes the backbone encoder as well as the decoder or predictor. $E$ refers to the drawing head.

## 3.1 Drawing Head: mask-then-draw

Given an image $x \in \mathbb{R}^{3 \times H \times W}$, we divide it into regular non-overlapping patches $\{p_1, p_2, ..., p_N\}$ as in ViT (Dosovitskiy et al., 2020), where $N$ denotes the number of patches. Following the sampling strategy (He et al., 2021), we randomly sample a subset without replacement and mask the remaining tokens. For simplicity, we denote the post-masking output as $x_M$. Existing MIM methods majorly map the observed signal $x_M$ to a latent representation using the backbone, and then reconstruct the tokenized (Bao et al., 2021) or original signal (He et al., 2021). We assign this job to a prepend drawing head. The goal is not to faithfully reconstruct the signal, but to model the uncertainty and produce diverse coarse drawings for the backbone. We follow noise injection in GANs (Brock et al., 2018; Feng et al., 2021b) to model the implicit probability function $p(x_M, z)$ of one-to-many mapping (i.e., given one masked image, there exist multiple outputs), where the noise vector is sampled from a unit Gaussian distribution $z \sim \mathcal{N}(0, 1)$ during training. Given different $z$, the drawing head $E_{draw}$ maps the masked input to diverse possible completions. Instead of using original signal $x$, we choose smoothed $\hat{x}$ to perform coarse reconstruction. The objective of this *mask-then-draw* process is to maximize the conditional likelihood of:

$$\arg\max_{\theta_{draw}} \mathbb{E}\left[\log p(\hat{x}|x_M, z)\right], \ z \sim \mathcal{N}(0, 1) \tag{1}$$

where $\theta_{draw}$ denotes the parameter of $E_{draw}$. Therefore, the drawing loss is defined as:

$$\mathcal{L}_{draw} = \|\hat{x} - E_{draw}(x_M, z)\|_2^2 \tag{2}$$

The loss function computes the mean squared error in the pixel space, between the drawing outputs and smoothed targets on masked patches. We illustrate the specific designs in the following.

**Head Architecture**  The drawing head consists of a lightweight structure: a token embedding stem, two transformer blocks as in ViT (Dosovitskiy et al., 2020), and a linear layer. It takes masked images and sampled noises as inputs, and outputs the coarse completions. Positional embeddings to all tokens are added. This incorporated module only appears during pre-training and incurs little computation overhead, as it is much smaller than the backbone or the asymmetric decoder in MAE (He et al., 2021).

**Noise Injection**  We modulate diverse outputs by injecting randomly sampled noise into the drawing head. Direct generating vanilla noise and concatenating with input is sometimes ineffective at modeling complex mappings (Feng et al., 2021b; Brock et al., 2018). We exploit a noise mapping network motivated by StyleGAN (Karras et al., 2019) to enhance randomness for stochastic variations. Our noise mapping network $f$ is a 3-layer MLP and translates the sampled noise vector $z$ to $w$, where $z$ and $w$ share the same dimensionality. Then the sampled and mapped noise vectors are injected into two transformer blocks in the drawing head respectively, as the extra token.

**Coarse Reconstruction**  MIM is challenging even for large transformers, due to the complexity of natural images. It is difficult for our lightweight drawing head to reconstruct original pixels that are sharp and realistic. As we target to handle diversity in this phase and don't count on generating perceptually-satisfying results, we employ a smoothed target $\hat{x}$ for training. Specifically, the simplified supervision is obtained by filtering the original images to reduce some high-frequency details, which is much easier for the drawing head. We adopt a Gaussian filter with fixed $\sigma$ and find this coarse reconstruction strategy is beneficial for stabilizing training.

### 3.2  Backbone Network: draw-then-predict

As part of the generation task is handed over to the drawing head, the backbone is able to focus more on understanding. We denote the coarse completion from the drawing head as $x_{draw}$. The backbone transformer $f_{net}$ extracts useful semantics from given $x_{draw}$ and a follow-up predictor $f_{pred}$ maps extracted features to the target modality, such as pixels (He et al., 2021; Xie et al., 2022) and HOG (Wei et al., 2021). During this *draw-then-predict* phase, we seek not to reconstruct pixels, but to perform feature prediction that is more beneficial for understanding the images. As the choice of pre-training objectives is flexible, we denote the potential feature extractor as $\mathbb{F}$. In practice, we obtain the targets $\mathbb{F}(x)$ from the original image $x$ instead of $\hat{x}$, for better fidelity and leaving space for generative capability. During pre-training, the backbone is optimized to minimize the negative cosine distance (i.e. equivalent to $\ell_2$-normalized mean squared error) between $\mathbb{F}(x)$ and the prediction:

$$\mathcal{L}_{pred} = -\frac{\mathbb{F}(x)}{\|\mathbb{F}(x)\|_2} \cdot \frac{f(x_{draw}))}{\|f(x_{draw})\|_2} \tag{3}$$

where $f(\cdot)$ denotes the backbone-to-predictor output for simplicity.

**Predictor Architecture**  The attached predictor plays an important role in learning good representation, as has been extensively explored (Grill et al., 2020; Chen & He, 2021; Caron et al., 2020). The decoder strategy in recent MIM frameworks (Xie et al., 2022; He et al., 2021) can also be regarded as an example of using predictors. Among the flexible choices of structure and capacity, we adopt a single linear layer as the predictor architecture, which we find both lightweight and effective. The predictor is only used during pre-training and introduces negligible computation overhead.

**Cross Prediction Targets**  During the *draw-then-predict* phase, our goal is to let the backbone understand the inherent contexts of coarsely completed images. Therefore, different prediction targets have a tendency to guide the network to distinctive levels of understanding. As the choice of target feature is flexible, we exploit the features from the widely adopted CLIP (Radford et al., 2021) model as pre-training objectives by default, based on the rich semantics within. Note that our proposed CrossMAE is general and could be compatible with various MIM prediction targets, such as HOG (Wei et al., 2021), discrete tokens (Bao et al., 2021) and

even the pixel reconstruction as in (Xie et al., 2022; He et al., 2021). We further perform an experimental analysis on potential targets in Section 4.4, as well as the ensembles of multiple targets.

In sum, the overall loss function of CrossMAE is written as:

$$\mathcal{L}(x, x_M) = \mathcal{L}_{draw}(\hat{x}; x_M, z) + \lambda \mathcal{L}_{pred}(x; x_{draw}) \qquad (4)$$

where $z$ is randomly sampled during training, $\hat{x}$ and $x_{draw}$ represent the smoothed objectives, and the output of drawing head when given $z$ and masked image $x_M$, respectively. The hyperparameter $\lambda$ balances the importance of *mask-then-draw* and *draw-then-predict*. For the masking strategy, we follow the random mask sampling of a 75% ratio as in MAE (He et al., 2021).

## 4 Experiments

In this section, we apply our CrossMAE pre-training scheme to a variety of tasks, including classification, detection, instance segmentation, semantic segmentation, and facial landmark detection. To further illustrate its effectiveness, we evaluate its data-efficiency in terms of both pre-training and fine-tuning. We also present comprehensive ablation studies to justify the contribution of each design and potential extensions.

### 4.1 Image Classification

**Implementation Details**  We evaluate our method on image classification using the popular ImageNet-1k (Deng et al., 2009) dataset, which consists of 1.3M images from 1000 categories. Specifically, we pre-train the backbone network on its training spilt and report the performance on the validation set, including both end-to-end fine-tuning and linear probing. Our pre-training setup generally follows the configurations in MAE (He et al., 2021) with AdamW (Loshchilov & Hutter, 2017) and cosine learning rate decay applied. The end-to-end fine-tuning and linear probing protocols are also kept consistent with those in (He et al., 2021; Bao et al., 2021). All models take the input size of 224×224 in the experiments of image classification.

**Comparison with State-of-the-arts**  We conduct comparisons with recent state-of-the-art MIM pre-training methods and report the top-1 accuracy in Table 1. Under the setting of end-to-end fine-tuning, where MIM-based frameworks dominate, CrossMAE outperforms previous *mask-then-predict* methods by a clear margin with only 300-epoch pre-training on the widely used ViT (Dosovitskiy et al., 2020) backbone. While CIM (Fang et al., 2022a) adopts a two-stage framework and conduct replaced token detection to train the network, we clearly beat its variant by 1%. Besides using ViT family, we also validate its effectiveness by adapting to the hierarchical Swin (Liu et al., 2021b) transformer architecture, where we simply append a linear layer to match the dimension of predictor due to Swin's structural design (i.e., downsampling). The fine-tuning results manifest the feasibility and semantic capability of CrossMAE. Linear probing is an established metric to evaluate the quality of learned representation, as we expect the representations large-scale pre-trained models to be plug-and-play. However, recent MIM methods often observe a performance gap between fine-tuning and linear probing. As the decoupled *mask-then-draw* and *drawing-then-predict* mitigate inputs and objectives inconsistency of the backbone, CrossMAE significantly surpasses traditional *mask-then-predict* pipelines in Table 1, without introducing contrastive learning (Zhou et al., 2021) and data augmentation (Chen & He, 2021).

### 4.2 Improvements on Downstream Tasks

To further validate the effectiveness of the proposed pre-training scheme, we evaluate it on downstream tasks including semantic segmentation, object detection and instance segmentation, as well as facial landmark detection. As detection and segmentation majorly focus on high-level understanding and semantic localization for natural objects, facial landmark detection involves face images and requires capturing fine-grained facial structures and local geometry. With different focus of these tasks, we intend to testify the generalization of our method. As our goal is not to perform extensive and heuristic feature selection, we adopt the consistent target of CLIP (Radford et al., 2021) in downstream task evaluations below. Additional details can be found in Section 4.4 and Appendix A.1.

| Method | Backbone | Target | PT Epoch | Fine-tune (%) | Linear (%) |
|---|---|---|---|---|---|
| BEiT[†] (Bao et al., 2021) | | DALL-E | 300 | 81.7 | 15.7 |
| CIM-RevDet (Fang et al., 2022a) | ViT-S/16 | DALL-E | 300 | 81.6 | - |
| CAE (Chen et al., 2022) | | Feature | 300 | 81.8 | 50.8 |
| CrossMAE (**Ours**) | ViT-S/16 | Feature[‡] | 300 | **82.1** | **66.8** |
| | | CLIP[*] | 300 | **82.5** | **72.9** |
| SimMIM (Xie et al., 2022) | Swin-B | Pixel | 100 | 83.5 | - |
| | | Pixel | 100 | **83.9** | **67.8** |
| CrossMAE (**Ours**) | Swin-B | Feature[‡] | 100 | **84.0** | **70.9** |
| | | CLIP[*] | 100 | **84.1** | **74.2** |
| BEiT (Bao et al., 2021) | | DALL-E | 800 | 83.2 | 56.7 |
| MAE (He et al., 2021) | | Pixel | 1600 | 83.6 | 67.8 |
| CAE (Chen et al., 2022) | ViT-B/16 | Feature | 800 | 83.6 | 68.3 |
| MaskFeat (Wei et al., 2021) | | HOG | 300 | 83.6 | - |
| SimMIM (Xie et al., 2022) | | Pixel | 800 | 83.8 | 56.7 |
| CrossMAE (**Ours**) | | Pixel | 300 | **83.8** | **70.9** |
| CrossMAE (**Ours**) | ViT-B/16 | Feature[‡] | 300 | **84.5** | **73.8** |
| CrossMAE (**Ours**) | | CLIP[*] | 300 | **84.7** | **76.3** |

Table 1: Comparisons with state-of-the-art MIM pre-training methods. **PT Epoch** denotes the number of pre-train epochs. **Fine-tune** and **Linear** report the top-1 accuracies on ImageNet validation set of fine-tuning and linear probing from the pre-trained model, respectively. ([†]: reported in (Chen et al., 2022), [‡]: adopts a moving average of the transformer encoder to provide the targets, as in (Grill et al., 2020), [*]: exploits additional knowledge.)

**Semantic Segmentation**  We evaluate our pre-training scheme on semantic segmentation on ADE20K (Zhou et al., 2019) dataset. To be specific, the backbone is firstly trained on ImageNet-1k using CrossMAE for 300 epochs, and then utilizes a task layer to perform semantic segmentation. We follow the consistent practice and configurations in BEiT (Bao et al., 2021) of ViT-B/16, with 160k fine-tuning iterations on 512×512 images. Similarly, multi-scale training and testing are not performed. In addition to using the task layer of UperNet (Xiao et al., 2018), we follow the practice in CIM (Zhou et al., 2021) to utilize a linear layer as decoder, which provides a more explicit comparison of the semantic quality of learned representation.

| Method | PT Epoch | mIoU |
|---|---|---|
| DINO (Caron et al., 2021) | 1600 | 43.0 |
| BEiT (Bao et al., 2021) | 300 | 43.2 |
| CIM (Fang et al., 2022a) | 300 | 43.5 |
| CrossMAE (**Ours**) | 300 | **44.6** |

Table 2: Comparison of semantic segmentation results (mIoU) on ADE20K dataset. All methods adopt ViT-B/16 as the backbone and a simple linear layer as decoder.

| Method | PT Epoch | mIoU |
|---|---|---|
| *using labeled data*: | | |
| DeiT (Touvron et al., 2021) | - | 45.6 |
| BEiT w Inter FT[†] | - | 47.7 |
| *without labeled data*: | | |
| BEiT (Bao et al., 2021) | 800 | 43.2 |
| PeCo (Dong et al., 2021) | 300 | 46.7 |
| MAE (He et al., 2021) | 1600 | 48.1 |
| CAE (Chen et al., 2022) | 800 | 48.8 |
| CrossMAE (**Ours**) | 300 | **50.4** |

Table 3: Results of semantic segmentation on ADE20K dataset, including methods pre-trained with and without labeled data. The adopted task layer is UperNet (Xiao et al., 2018). ([†]: with intermediate fine-tuning in (Bao et al., 2021))

From Table 2, we find our method outperforms several pre-training schemes that cover contrastive or MIM objectives. While the difference among the three representatives is marginal, our method boosts the backbone by a large margin. Considering the simplicity of the adopted linear decoder, the comparison verifies the nice quality and rich semantics learned by CrossMAE directly. When using a more complex task layer (Xiao et al., 2018) as shown in Table 3, our method significantly surpasses MIM unsupervised or supervised pre-training counterparts. The results further indicate the importance of proper pre-training.

**Object Detection and Instance Segmentation**   We study the performance of CrossMAE pre-trained model on object detection and instance segmentation using the COCO 2017 (Lin et al., 2014) dataset. Specifically, we adopt the popular Mask R-CNN (He et al., 2017) framework and employ the pre-trained transformer encoder from CrossMAE as the backbone for the detector. After obtaining the ImageNet pre-trained weights, we fine-tune the model on the `train` split and report the performance on the `val` split. To encode multi-scale information into the detection pipeline, we integrate FPN (Lin et al., 2017) into the backbone following the setup in (Zhou et al., 2021; He et al., 2021). The performance from different pre-training schemes and backbone architectures are reported in Table 4.

| Method | Backbone | PT Epoch | FT Epoch | $AP^{box}$ | $AP^{mask}$ |
|---|---|---|---|---|---|
| *with supervised pre-training*: | | | | | |
| DeiT Scratch (Li et al., 2021a; Touvron et al., 2021) | ViT-B | 300 | 50 | 47.9 | 42.9 |
| Swin (Liu et al., 2021b) | Swin-S | 300 | 36 | 48.5 | 43.3 |
| *with MIM pre-training*: | | | | | |
| BEiT (Bao et al., 2021) | ViT-B | 300 | 12 | 42.6 | 38.8 |
| PeCo (Dong et al., 2021) | ViT-B | 800 | 12 | 44.9 | 40.4 |
| CAE (Chen et al., 2022) | ViT-B | 800 | 36 | 49.2 | 43.3 |
| BEiT$^{\dagger}$ (Bao et al., 2021; Li et al., 2021a) | ViT-B | 800 | 100 | 49.8 | 44.4 |
| MAE (He et al., 2021; Li et al., 2021a) | ViT-B | 1600 | 100 | 50.3 | 44.9 |
| CrossMAE (**Ours**) | ViT-B | 300 | 36 | **50.9** | **45.2** |

Table 4: Object detection and instance segmentation performance on COCO. **PT Epoch** and **FT Epoch** denote the number of pre-train and fine-tuning epochs. The methods included are pre-trained either with labels or under MIM schemes. FPN is included in the backbone for capturing hierarchical information. ([†]: reported in (Li et al., 2021a))

As shown in Table 4, MIM pre-training methods clearly beat the previous supervised pre-training on the final results. However, the performance drop on BEiT (Bao et al., 2021; Li et al., 2021a) under insufficient pre-training and fine-tuning epochs is quite obvious. In contrast, our method easily surpasses existing MIM approaches under a much shorter pre-training and fine-tuning scheduler. Note that our work targets at providing a generic pre-training scheme and exploits the popular plain ViT and FPN architecture when evaluating downstream tasks. As several concurrent works such as ViTDet (Li et al., 2022) and MIMDet (Fang et al., 2022b) investigate designs on the backbone and hierarchical connection, we believe our contributions are mutually beneficial for obtaining better downstream performance.

**Facial Landmark Detection**   In contrast to high-level understanding with classification, detection, and segmentation, the requirement of facial landmark detection focuses more on extracting accurate facial geometry and handling variations or distortions. To validate the generalization of CrossMAE, we test facial landmark detection on WFLW (Wu et al., 2018b). The challenging dataset consists of 98 manually annotated landmarks and includes large variations in expression, pose and occlusion. Due to the domain gap between face and natural images, we pre-train the backbone on WFLW instead of ImageNet. Note that during pre-training, we only train the model on the training split and use no extra data or label. The model is then fine-tuned on the training set using landmark annotations. Similar to semantic segmentation, we use ViT-B as our backbone and adopt the UperNet (Xiao et al., 2018) structure as the task layer to output heatmaps as in (Zheng et al., 2022).

| Method | Mean Error (%) | | | | | | | FR | AUC |
|---|---|---|---|---|---|---|---|---|---|
| | Full | Pose | Expression | Illumination | Make-up | Occlusion | Blur | | |
| CFSS (Zhu et al., 2015) | 9.07 | 21.36 | 10.09 | 8.3 | 8.74 | 11.76 | 9.96 | 20.56 | 36.6 |
| LAB (Wu et al., 2018a) | 5.27 | 10.24 | 5.51 | 5.23 | 5.15 | 6.79 | 6.32 | 7.56 | 53.2 |
| SAN (Dong et al., 2018) | 5.22 | 10.39 | 5.71 | 5.19 | 5.49 | 6.83 | 5.8 | 6.32 | 53.6 |
| WING (Feng et al., 2018) | 5.11 | 8.75 | 5.36 | 4.93 | 5.41 | 6.37 | 5.81 | 6.0 | 55.0 |
| AVS (Qian et al., 2019) | 4.76 | 8.21 | 5.14 | 4.51 | 5.00 | 5.76 | 5.43 | 5.24 | 54.6 |
| Scratch (Zheng et al., 2022) | 4.8 | 8.78 | 5.09 | 4.74 | 4.99 | 6.01 | 5.35 | 5.72 | 54.5 |
| CrossMAE (**Ours**) | **4.21** | **6.99** | **4.42** | **4.13** | **4.1** | **5.06** | **4.85** | **2.8** | **59.32** |

Table 5: Facial landmark detection results on WFLW. **Mean Error** is normalized by the inter-ocular distance. **FR** denote Failure Rate (%). We follow evaluation protocols used in (Wu et al., 2018b). Scratch refers to the same architecture trained from scratch with randomly initialized weights, as reported in (Zheng et al., 2022).

The test set is partitioned into 6 subsets based on the challenging attributes of pose, expression, illumination, make-up, occlusion, and blur. From Table 5, we can see that the model trained from scratch already shows competitive results on several subsets. By utilizing CrossMAE pre-training, the model achieves 4.21% mean error, outperforms the state-of-the-art method (Qian et al., 2019) which leverages synthetic data by a large margin. The results demonstrate the effectiveness and generalization of CrossMAE on different vision tasks. Meanwhile, the consistent improvement on all subsets show that the model with pre-training is less sensitive towards extreme structure and appearance variations. The results indicate the potentials on improving visual invariance from our diverse modeling strategy.

### 4.3 Efficiency under Limited Data Scenarios

As the goal of modeling diverse generation and understanding with CrossMAE is to learn a more generalizable feature presentation beyond memorization, we verity its data efficiency under two scenarios: limited data for pre-training, and limited labeled data during fine-tuning.

| Method | PT PCT (%) | |
|---|---|---|
| | 1% | 10% |
| Supervised | 71.6 | 75 |
| DINO (Caron et al., 2021) | 70.1 | 73.1 |
| BEiT (Bao et al., 2021) | 74.1 | 74.5 |
| SplitMask (El-Nouby et al., 2021) | 74.8 | 75.4 |
| CrossMAE (**Ours**) | **77.6** | **78.1** |

Table 6: Transfer performance of different pre-training methods to iNaturalist-2019. **PT PCT** denotes the percentage of ImageNet data used for pre-training, under the same number of iterations.

| Method | Arch | FT PCT (%) | |
|---|---|---|---|
| | | 1% | 10% |
| SimCLRv2 (Chen et al., 2020c) | Res50 | 57.9 | 68.1 |
| BYOL (Grill et al., 2020) | Res50 | 53.2 | 68.8 |
| SwAV (Caron et al., 2020) | Res50 | 53.9 | 70.2 |
| ReLICv2 (Tomasev et al., 2022) | Res50 | 58.1 | 72.4 |
| DINO (Caron et al., 2021) | ViT-S/16 | 60.3 | 74.3 |
| iBOT (Zhou et al., 2021) | ViT-S/16 | 61.9 | 75.1 |
| CrossMAE (**Ours**) | ViT-S/16 | **64.1** | **76.5** |

Table 7: Accuracy of different models after fine-tuning. **FT PCT** denotes the label fraction used in fine-tuning.

**Data Efficiency on Pre-training** We study the pre-training data efficiency following the settings in SplitMask (El-Nouby et al., 2021), where the model is pre-trained with a reduced number of ImageNet samples and evaluated with its transfer performance on iNaturalists (Van Horn et al., 2018). From Table 6, we can find that MIM based methods perform consistently and are much better than supervised or contrastive methods when pre-trained in low-data regimes. This difference between 1% and 10% suggests the good data utilization of MIM pre-training, while CrossMAE significantly outperforms other MIM methods.

**Label Efficiency on Fine-tuning** A common situation to use pre-trained model is when limited labeled data are given. We show the fine-tuning efficiency by comparing with methods that follow the *unsupervised pre-training, then supervised fine-tune* paradigm as in iBOT (Zhou et al., 2021) and SimCLRv2 (Chen et al., 2020c). Table 7 shows the excellent label efficiency of our method, verifying its potentials for general usage.

## 4.4 Further Analysis

To better understand different components in CrossMAE, we conduct ablation studies on the structural designs and dissect the prediction objectives. We first degrade the overall pipeline to baseline with the computation graph in Figure 2 (a). Then we gradually assemble the drawing head, noise modeling, and feature prediction to the framework. These variants are pre-trained with the consistent setting in Section 4.1. We also provide additional study to understand the prediction process.

| Variant ↓ | Implementation | FT(%) | Lin(%) |
|---|---|---|---|
| Baseline | - | 83.0 | 54.3 |
| + Draw | Original $x$ | n/a | n/a |
| | **Coarse $\hat{x}$** | 83.3 | **71.2** |
| + Noise | Concatenation | 83.2 | 69.7 |
| | **Mapping Network $f$** | **83.8** | 70.9 |
| + Prediction | without Predictor | 84.1 | 71.5 |
| | **with Predictor $f_{pred}$** | **84.7** | **76.3** |

Table 8: Ablations on different modules and implementations. **FT**, **Lin** respectively denote the fine-tuning and linear probing results. We sequentially assemble the modules in ↓.

| Type | Target | Acc | mIoU | AP |
|---|---|---|---|---|
| Single | Pixel | 83.1 | 46.7 | 48.9 |
| | VGG | 82.9 | 46.3 | 49.1 |
| | Canny | 82.2 | 47.5 | 48.6 |
| | HOG | 83.4 | 48.5 | 48.9 |
| | EMA Feat | 83.6 | **50.3** | **49.2** |
| | CLIP | **83.9** | 50.1 | 49.3 |
| Multi | CLIP + HOG | 83.3 | **50.4** | 49.1 |

Table 9: Comparison of different targets $\mathbb{F}$. **Acc**, **mIoU** and **AP** denote fine-tuned accuracy on ImageNet, mIoU on ADE20k segmentation and AP box on COCO detection.

**Effect of Drawing Head** Reconstructing complex pixels from masked inputs is challenging for the lightweight drawing head. We find that the loss easily diverges (NaN) under the original target $x$. The coarse $\hat{x}$ stabilizes training and brings a *coarse-to-fine* transition. As shown in Table 8, the drawing head improves linear probing by 15%. This incorporated design provides coarse but completed images to the backbone, thus making the backbone focus more on *understanding* in pre-training.

**Effect on Noise Modeling** Noise modeling plays an important role in producing diverse completions during pre-training. From the comparison in Table 8, the fine-tuning results are improved for about 1% over the baseline and the performance of linear probing is roughly remained. We conjecture that the model strengthens its capability and visual invariance on feature prediction, which is further exploited when being fine-tuned. We also compare different ways of noise injection and empirically find that, by naively concatenating sampled noise to the input, the method performs poorly and contributes subtle diversity in drawing head. This observation shows the effectiveness of the noise mapping network $f$ on handling stochastic variations. This curious mechanism has also been investigated theoretically in GAN inversion (Feng et al., 2021b) and image generation (Alharbi & Wonka, 2020).

In addition to the accuracy improvement, we provide another perspective of noise modeling as sophisticated online augmentation on masked regions. Following the setting in Table 7, we validate its benefit on low-data fine-tuning regime, by removing the incurred noise. From Table 10, the noise modeling consistently helps performance and improves the accuracy more under limited data. These results can further demonstrate the boosted generalization.

| Method | Architecture | FT PCT (%) | |
|---|---|---|---|
| | | 1% | 10% |
| wo Noise | ViT-S/16 | 62.1 | 75.3 |
| **w Noise** | ViT-S/16 | **64.1** | **76.5** |

Table 10: Effect with noise after fine-tuning. **FT PCT** denotes the label fraction used in fine-tuning.

| Method | Params | mCE (↓) |
|---|---|---|
| Swin-B Liu et al. (2021b) | 87.8M | 54.4 |
| Ours wo Noise | 86.8M | 51.7 |
| **Ours w Noise** | 86.8M | **48.9** |

Table 11: Normalized mean corruption error **mCE** on ImageNet-C, which measures the robustness towards corruptions.

As shown in Figure 3, the drawing head is capable of completing varying texture on masked regions. We further validate whether our noise modeling design upgrades the robustness of vision transformers towards natural corruptions on ImageNet-C Hendrycks & Dietterich (2019). Though transformers have demonstrated dominance against corruptions compared to CNNs Zhou et al. (2022); Bhojanapalli et al. (2021), the incurred diversity still enhances the robustness of the baseline structure as shown in Table 11, which quantifies its benefits on extracting visual invariance.

**Understanding Feature Prediction**   To better understand, we conduct pre-training with different prediction targets $\mathbb{F}$. We adopt several popular feature extractors in computer vision and study their fine-tuned performance on two different tasks: image classification and semantic segmentation. We pre-train each variant in 100 epochs and follow other settings consistently. As shown in Table 9, while perceptual loss (Johnson et al., 2016) shows advantages over pixelwise $\ell_2$ loss on image generation quality, adopting VGG (Simonyan & Zisserman, 2014) as $\mathbb{F}$ shows degraded scores on both tasks. Although using Canny edge detector (Canny, 1986) leads to 1% drop on classification, the improvement on ADE20k over pixels implies the connection between segmentation and edge awareness. We also exploit HOG (Dalal & Triggs, 2005) following MaskFeat (Wei et al., 2021) to our framework and see a clear boost. Beyond that, the vision branch of CLIP (Radford et al., 2021) exhibits an advantage especially on classification. The reason might be the multi-modal nature of CLIP, that inherently distills rich semantics to our backbone during pre-training. We also assemble two separate predictors to jointly learn from CLIP and HOG. Despite the dropped accuracy, this variant leads to a better segmentation model. Unlike classification in Table 9, we don't observe a significant improvement on downstream tasks with CLIP features. While not seeking to find an optimal prediction target, we can conclude a crucial takeaway from the comparisons: *it's beneficial to design our MIM pre-training objectives that connect to some desired properties in downstream tasks*. We regard investigating the detailed mutual relations as our future work.

### 4.5   Additional Elaboration on the Effectiveness and Efficiency

To further allow a consistent comparison, we report the wall-clock pre-training time on the same platforms to better illustrate this advantage over the baseline MAE (He et al., 2021). The hardware we adopted contains 8 V100 GPUs. From Table 12, we can clearly see the performance improvements under the same pre-training time, even with raw pixels.

| Method | Targets | PT Epochs | Wall-clock Time | Fine-tune (%) | Linear (%) |
|---|---|---|---|---|---|
| MAE | Pixel | 300 | 78h | 83.1 | 61.5 |
| MAE | Pixel | 1600 | 464h | 83.6 | 67.8 |
| MAE | Pixel | 410 | 107h | 83.2 | 62.4 |
| **CrossMAE** | Pixel | 300 | 107h | **83.8** | **70.9** |
| MAE | Pixel | 530 | 138h | 83.2 | 63.7 |
| **CrossMAE** | Features | 300 | 138h | **84.5** | **73.8** |

Table 12: Comparisons on pre-training effectiveness and efficiency. **Wall-clock Time** denotes the pre-training hours. **Fine-tune** and **Linear** report fine-tuned and linear accuracies on ImageNet-1k.

We also emphasize that feature prediction is crucial in the design of CrossMAE. To avoid the extra knowledge from CLIP features, we adopt a BYOL-style (Grill et al., 2020) momentum encoder from the online network and utilize its prediction as targets. By ensuring no additional information is leveraged, we can see the fine-tuning performance is improved significantly from 83.8 to 84.4, which shows the importance and benefits of decoupled feature prediction. We also show the MAE baseline with the same pre-training time for comparison.

**Details on Coarse Reconstruction**   We present more details on the coarse reconstruction. As mentioned, we employ a smoothed target $\hat{x}$ to reduce the training difficulty of *mask-then-draw*. Due to the simple structure of the drawing head and complexity of natural images, we do not count on providing realistic completions to

the backbone transformer. As have been analyzed in (Bhojanapalli et al., 2021; Zhou et al., 2022), vision transformers show intriguing properties and robustness towards corruptions over CNN counterparts. We also find that in *draw-then-predict*, the backbone transformer can still be well trained even with significant blur and distortions on the coarse completions from *mask-then-draw*. These nice properties of vision transformers enable us to apply the coarse reconstruction objective in training, which significantly reduces training difficulty.

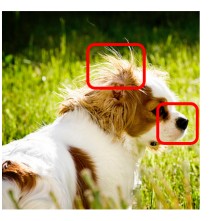 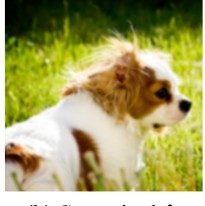 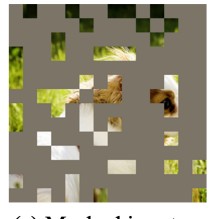 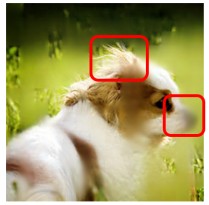 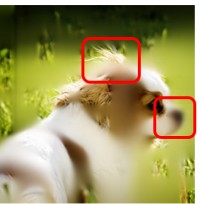

(a) Original $x$     (b) Smoothed $\hat{x}$     (c) Masked input $x_M$     (d) Two sampled $x_{draw}$

Figure 3: Illustration of the *mask-then-draw* process. We visualize an example image $x$ and the smoothed target for the drawing head in (a) and (b). Then a randomly masked $x_M$ is provided as the input to the drawing head. Since our method models one-to-many mapping in the drawing step, we sample two completions $x_{draw}$ from different $z$. We can clearly see the difference on reconstructed regions such as the nose and the dog hair. (**Zoom in for better details**)

In practice, we adopt a Gaussian kernel with $\sigma = 10$ to smooth the original images and obtain the coarse objective $\hat{x}$. To better understand the *mask-then-draw* process in our framework, we provide visualizations in Figure 3. We can see that $\hat{x}$ removes some details from the original image, therefore reducing the difficulty of reconstruction. To show the diverse completions from our drawing head, we also visualize the masked input and two sampled completions from different injected noise $z$. The difference in two completions, such as variations in the dog's nose, hair and texture, demonstrates the one-to-many mapping from our designs.

## 5    Conclusion

In this work, we dissect existing MIM pipelines by decoupling its generation and understanding steps. We demonstrate the strong performance brought by the proposed pre-training method called CrossMAE, through extensive experimental results. Our study suggests the cruciality of defining a proper objective during pre-training and also reveals both the computational and data inefficiency of naive pixel reconstruction on natural images. Therefore, we hope our simple but effective approach will serve as a solid and general pre-training baseline and inspire future studies in visual representation learning.

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

# A  Appendix

The content of the appendix is organized as follows:

- Appendix A.1 provides additional ablations about the structure of the drawing head and predictor, as well as balancing loss weights and target selection on downstream tasks.

- Appendix A.2 provides further discussion on contrastive and MIM-based methods.

- Appendix A.3 presents more experimental settings and hyper-parameters for experiments on ImageNet and COCO.

## A.1  Additional Ablation Studies

**Structure of Drawing Head**  The drawing head plays an important role in our framework. As we have analyzed the effect on this design as well as noise modeling, we also study how different structures of the drawing head influence the results. Since a stronger head might provide completions with better quality, we increase the number of transformer blocks in $E_{draw}$ by several times and experiment with these modified designs respectively. In addition, we replace the light-weight transformer head with a U-Net architecture to testify whether CNNs apply to the drawing head.

| Architecture | Blocks | Costs | Fine-tune (%) |
|---|---|---|---|
| | 2 | 1× | 84.7 |
| Transformer | 4 | 1.8× | 84.8 |
| | 6 | 2.6× | 84.3 |
| | 8 | 3.5× | 84.0 |
| CNN | U-Net (Ronneberger et al., 2015) | 1.7× | 84.1 |

Table 13: Ablations on different structures of the drawing head. **Blocks** refer to the numbers of transformer blocks when using transformer-based heads. When it comes to CNN-head, it denotes the network design. **Costs** denotes the computation costs compared to our baseline method that adopts a 2-block structure as shown in the first row. **Fine-tune** reports fine-tuned accuracies on ImageNet-1k.

From Table 13, we can see that enlarging the capacity of $E_{draw}$ does not necessarily mean better results. While it witnesses marginal improvements initially, too heavier drawing head degrades the performance clearly. The variance probably comes from the increased training difficulty by these designs. Meanwhile, we also demonstrate the possibility of using CNNs as our drawing head. Although current results are not competitive with transformer-based heads, it out-performs the baseline without *mask-then-draw* such as MAE (He et al., 2021) process by 0.5%. The results indicate the feasibility of different head structures, which can be further explored.

**Structure of Predictor**  Table 14 ablates the effect from different predictor, where we experiment with a single linear layer, the 2-layer predictor MLP with BN (Ioffe & Szegedy, 2015) applied internally as in SimSiam (Chen & He, 2021), and the MAE decoder that has 8 blocks and a width of 512-d. From the comparisons in Table 14, we find that our simple single-layer predictor performs on par with other more complex designs. Considering the computational overhead brought by heavier predictors, we exploit the simple choice of a linear layer as our predictor.

| Predictor Architecture | Costs | Fine-tune (%) |
|---|---|---|
| Linear | 1× | 84.7 |
| 2-layer MLP with BN | 1.2× | 84.6 |
| Transformer Decoder | 1.8× | 84.7 |

Table 14: Ablations on architectures of the predictor. In **Predictor Architecture**, we compare our design with predictors or decoders in (Chen & He, 2021; He et al., 2021). **Fine-tune** reports fine-tuned accuracies on ImageNet-1k.

**Balancing Loss Weights and $\sigma$**  Through our experiments, we find that the transfer performance of the pre-trained model is robust towards varying $\lambda$ and $\sigma$, which balances the weights of $\mathcal{L}_{draw}$ and $\mathcal{L}_{draw}$. However, too large $\lambda$ could lead to numerical instability. Due to the difficulty of *draw-then-predict* that occasionally leads to higher losses, for instance, 0.65 for $\mathcal{L}_{pred}$ versus 0.09 for $\mathcal{L}_{draw}$, we set $\lambda$ to 0.1 and $\sigma$ to 10 in our experiments unless otherwise stated.

**Target Selection on Downstream tasks.**  During our experiments, CLIP (Radford et al., 2021) feature is not optimal for different downstream tasks. Intuitively, we expect the CLIP target to work well on ImageNet classification with its rich semantics. However, we find that on other downstream tasks, it occasionally performs worse than other targets: for example, on facial landmark detection, the perceptual (Johnson et al., 2016) feature out-performs CLIP with a normalized mean error (NME) of 4.07, which is clearly lower than the 4.21 we reported using CLIP; on object detection and instance segmentation, the exponential moving average encoder serves as a competitive scheme. Note that selecting the best target for different downstream tasks is outside the context of CrossMAE, where we don't intend to perform extensive and

| $\lambda$ | $\sigma$ | **Accuracy** |
|---|---|---|
| | 1 | 84.2 |
| 0.1 | 5 | 84.6 |
| | 10 | 84.7 |
| | 20 | 84.8 |
| 0.4 | | 84.7 |
| 1.0 | 10 | 84.5 |
| 10 | | diverge |

Table 15: The final fine-tuned accuracy given different loss weights $\lambda$ and $\sigma$.

heuristic feature selection. Therefore, we choose a relatively consistent target of CLIP (Radford et al., 2021) in downstream task evaluations, to better convey the general idea of CrossMAE: decoupling low-level generation and high-level understanding. As potential relations exist in various targets and downstream tasks, we regard investigating target selection as our future work.

### A.2 Discussion on Contrastive and MIM-based methods

We provide further discussion on existing contrastive learning and recent MIM-based methods. Serving as the primary direction in self-supervised learning, instance discrimination frameworks (Wu et al., 2018c; Chen et al., 2020b;d) has demonstrated strong performance in recent years. With the development of vision transformer, we've also witnessed a booming growth in learning stronger representations using these structures. In parallel with contrastive-based methods such as MoCov3 (Chen et al., 2021) and SwAV (Caron et al., 2020), another line of research in masked image modeling (MIM) recently caught attention (Bao et al., 2021; He et al., 2021). As there exists distinctive difference between these two methods, we provide additional discussion on them.

**Advantage of Contrastive Learning** Contrastive learning methods usually report a good kNN and linear accuracy. Based on the their training objectives, the contrastive loss focuses on aligning instance-level feature and separating the instance-wise distance. These characteristics directly contribute to better linear separability and hence lead to strong linear accuracy. Meanwhile, techniques such as multi-crop augmentation (Caron et al., 2020), which increase the number of views, benefit comparison between different views of images. Therefore, we expect a satisfying linear probing result and the capability of frozen feature extraction from these contrastive pre-trained models.

**Advantage of Masked Image Modeling** Without the constraints such as feature alignment and uniformity (Wang & Isola, 2020) from contrastive loss, MIM focuses on reconstructing the masked regions based on other visible regions. As expected, MIM pre-training often leads to a low linear score and requires a necessary fine-tuning process. Meanwhile, with the vast development of MIM-based methods (Li et al., 2021a; He et al., 2021), people find that fine-tuned MIM models produce surprising results, despite poor linear performance. MIM's good generalization towards different downstream tasks is a charming advantage, which significantly outperforms previous supervised and contrastive pre-training. We've also demonstrated these benefits in our experiments.

As the debate over contrastive and MIM-based methods still goes on, our efforts aim at finding a better MIM pre-training paradigm, as well as narrowing the pretraining-finetuning gap between contrastive and MIM methods. We believe we've demonstrated in the experiments that CrossMAE is a powerful and generalizable MIM pre-training scheme. Also, it's important to discover a versatile prediction target that adapts to different downstream tasks. In order to close this gap, another interesting future direction is to leverage both advantages in contrastive and MIM methods.

| Config | Value |
|---|---|
| Optimizer | AdamW |
| Base lr | 1.5e-4 |
| weight decay | 0.05 |
| optimizer momentum | $\beta_1, \beta_2 = 0.9, 0.95$ |
| batch size | 512 |
| Learning rate schedule | cosine decay |
| Warmup epochs | 20 |
| Augmentation | RandomResizedCrop |

Table 16: **Pre-training setting.**

| Config | Value |
|---|---|
| Optimizer | SGD |
| Base lr | 0.1 |
| Weight decay | 0 |
| Optimizer momentum | 0.9 |
| Batch size | 512 |
| Learning rate schedule | cosine decay |
| Warmup epochs | 10 |
| Training epochs | 90 |
| Augmentation | RandomResizedCrop |

Table 17: **Linear probing setting.**

### A.3 Experimental Settings and Hyper-parameters

**ImageNet Experiments** For the experiments on ImageNet, we follow the most standard settings and hyper-parameters in MAE (He et al., 2021) when using ViT (Dosovitskiy et al., 2020) as the backbone. When evaluating our method on Swin Transformer (Liu et al., 2021b), we flatten and downsample the features from the prediction target $\mathbb{F}$ using an additional linear layer, to match the output dimension of Swin. Each experiment is performed on 8 Tesla V100 GPUs. During pre-training, we follow MAE (He et al., 2021) to use the Xavier initialization (Glorot & Bengio, 2010) and choose not to adopt color jittering and drop path. Our fine-tuning regimes follow BEiT (Bao et al., 2021), where we find the layer-wise learning rate decay plays an important part. For linear probing, the protocols in MoCov3 (Chen et al., 2021) and MAE (He et al., 2021) are adopted.

| Config | Value |
|---|---|
| Optimizer | AdamW |
| Base lr | 1e-3 |
| Weight decay | 0.05 |
| Optimizer momentum | $\beta_1, \beta_2 = 0.9, 0.999$ |
| Layer-wise lr decay | 0.65 |
| Batch size | 1024 |
| Learning rate schedule | cosine decay |
| Warmup epochs | 5 |
| Training epochs | 100 |
| Augmentation | RandAug (9, 0.5) Cubuk et al. (2020) |
| Label smoothing Szegedy et al. (2016) | 0.1 |
| Mixup Zhang et al. (2017) | 0.8 |
| Cutmix (Yun et al., 2019) | 1.0 |
| Drop path Huang et al. (2016) | 0.1 |

Table 18: **Pre-training setting.**

**COCO Experiments** The Mask R-CNN (He et al., 2017) framework is exploited as our object detection and instance segmentation head. As ViT is adopted as the backbone, we integrate the FPN (Lin et al., 2017) backbone following (Zhou et al., 2021), where the transformer blocks are divided into 4 parts and the intermediate feature maps are applied with convolutions to obtain multi-scale features. All of our models are trained with the $3\times$ schedule. The AdamW (Loshchilov & Hutter, 2017) optimizer with an initial learning rate of 6e-5 and a weight decay of 0.05 is adopted.

