# OpenReview forum: "CrossMAE: Decoupled Masked Image Modeling with Cross Prediction"
_TMLR — Rejected by TMLR_

### Review · Reviewer_zo7M · 2022-10-04

**Summary Of Contributions:**

This paper introduces a new loss function for masked image modeling training. The new loss function includes two parts: 1) a loss that minimizes blurred reconstruction error of masked inputs, and 2) a loss that ensures that the noisy reconstruction and the original image produce similar CLIP features. The authors test their model on image classification and other downstream computer vision tasks.

**Broader Impact Concerns:**

None.

**Requested Changes:**

### Critical Changes

The following changes would be necessary for me to consider acceptance of this paper:

1. **Address the unscientific language throughout the paper.** (See weaknesses section.)
2. **Compare against baselines (e.g. MaskFeat) that use CLIP features.** This will make for a more fair comparison with CrossMAE, and it will also demonstrate whether the use of CLIP features must be combined with CrossMAE or whether it is a useful trick that can boost the performance of any MIM method. (See weaknesses section)
3. **Properly test the role of the added noise $z$.** (See weaknesses section.)
4. **Either change the motivation of CrossMAE** (it cannot be about "learning more disentangled features" if there is no actual investigation of the features) **or include an actual investigation of the learned features.** (See weaknesses section.)

### Minor Changes

- You refer to DINO as a contrastive method in the Sections 1 and 2. However, DINO is not contrastive.
- "A trivial solution is to memorize the training datapoints (Nagarajan et al., 2018), thus generalizing poorly." (Page 3) This is not necessarily true. Memorization has been shown to be beneficial in numerous contexts. See for example: https://arxiv.org/abs/2008.03703 and https://dl.acm.org/doi/abs/10.1145/3406325.3451131.
- "...As the disentangled mask-then-draw and drawing-then-predict mitigate inputs and objectives inconsistency of the backbone.” (Conclusion) You have not demonstrated this. You have only demonstrated performance, but you have not demonstrated that this is due to fixing "inconsistencies" in the objective.
- The ablation study needs more information. What is the "single-level" noise concatenation that you test in this section?
- Equation 3 is rather confusing. “...where p(·) denotes the backbone-to-predictor output for simplicity.” What does this mean? I am assuming that p() is not measuring a probability, and in that case you should strongly consider using different notation.

### Typos

- Page 1 paragraph 2: acronyms like BEiT, MAE, etc should be defined
- Figure 1 caption: “one-the-many” should be “one-to-many”

**Strengths And Weaknesses:**

## Strengths

- This method achieves promising results on image classification and downstream tasks
- This paper investigates using CLIP features in the MIM training objective, rather than relying on raw pixels or HOG features. The use of CLIP features appears to be promising.
- MIM is timely and of high interest to the community.

## Weaknesses

### Motivation for CrossMAE is Not Addressed or Tested

In the introduction, CrossMAE is motivated to address the limitations of existing MIM methods on downstream tasks. These limitations are conjectured to be the inability to "bring alignment and uniformity on features" and the inability to "learn rich semantic features." However, the central hypothesis of this work (that CrossMAE addresses these limitations) is never explicitly tested.

While CrossMAE does appear to get better performance on downstream tasks than other MIM methods (Table 2), the performance improvement appears to be roughly the same order of magnitude as CrossMAE’s improvement on standard image classification tasks (Table 1). Therefore, an alternative explanation to the success of CrossMAE is that its superior performance is strictly due to it learning better image classification representations, rather than it taking a step towards solving the fundamental limitations described in the Introduction. (Moreover, it is likely that this performance improvement can be largely explained by using CLIP features in the training process, which may give CrossMAE an unfair advantage over baseline methods; see below.)

If you wish to motivate your method through the limitations of existing MIM methods (the inability to “bring alignment and uniformity on features” and the inability to “learn rich semantic features”), this must be backed up by analyses of the features that demonstrate that CrossMAE is learning significantly different features than other MIM methods. Otherwise, this motivation and explanation for the success of CrossMAE is unscientific.

(Moreover, the second paragraph on page 2, which introduces this motivation, is very confusing. “The generative supervisory signals bring inconsistency to pre-training and downstream tasks.” This paragraph is confusing: I am unclear what the limitations of MIM methods actually are. The choice of wording here is vague (e.g. “alignment and uniformity,” “rich semantic features,” etc.) and - though it is possible that someone more up-to-date on the pre-training literature may understand these concepts, this lack of specificity is alienating to those without intimate knowledge of these works.)

### A Possible Unfair Comparison?

A primary difference between CrossMAE and the other baseline methods is that CrossMAE training involves matching CLIP features between the reconstructed and original images. It is very likely therefore that the use of CLIP (which is a learned representation) has a strong impact on the performance of CrossMAE. This is indeed confirmed by the ablation study in Table 9.

However, all of the comparison methods (to the best of my knowledge) do not use any additional learned representations in the training procedure. (For example MaskFeat, SimMIM, and MAE only use raw pixels or HOG features.) This gives CrossMAE an advantage over these methods, and so the success of CrossMAE over these methods cannot be solely attributed to any of the motivating intuitions that the authors provide (i.e. disentangled objectives leading to richer features).

### The Role of the Added Noise is Suspect

The added noise $z$ is supposed to "model the uncertainty and produce diverse coarse drawings for the backbone." First, this language is vague an imprecise (what does it mean to "model uncertainty?" Moreover, you never attempt to quantify "diversity?"). More critically, it is unclear what the noise is actually doing in the loss function.

First, the loss function in Eq. 1 does not "model uncertainty." You are optimizing a conditional expected log likelihood, which will be maximized when there is little uncertainty. The loss function will aim to make x_M invariant to z: this is not the same as “modeling uncertainty." Second, there is nothing in the loss function that is forcing E_draw to pay attention to z. Eq (1) and (2) will likely be maximized when E_draw ignores z.

From this high level analysis of Eq (1) and (2), it appears that z does not actually contribute anything to the method!

### Unscientific Language and Descriptions

There are many examples where the authors are using imprecise and unscientific language to describe phenomena. This is problematic, especially in light of the previous weakness, since it makes it difficult to actually dissect and understand how the proposed method actually improves upon ViT. Some examples:

- "However, due to lack of enough inductive bias, vision transformers often yield lowered performance on mid-sized datasets" (Page 1) How do you quantify "lack of enough inductive bias?" Also, do you have a citation for this?")
- "Whereas the imperfect completions share ambiguity and diversity, the backbone plays a different role in understanding their potential contexts." (Page 2) What does it mean to "share ambiguity and diversity?" How do you quantify ambiguity and diversity?
- "While contrastive method constrains the feature space and brings nice properties (Wang & Isola, 2020)..." (Page 3) What are "nice properties"? And again, how are these "nice properties" quantified?
- "...where the backbone focuses more on understanding the contents and involves computing the visual invariance" (Page 3) How do you quantify this? There are no discussions about computing visual invariance at all throughout this paper, nor do you test this at all.
- "...our goal is to model the diversity during pre-training." (Page 3). What does it mean to "model diversity"? Again, how do you quantify this?
- "It is overwhelming and inefficient for our lightweight drawing head to reconstruct original pixels that are sharp and realistic." (Page 5). What does it mean for a drawing head to be "overwhelmed?"
- "Our study suggests the cruciality of defining a proper objective during pre-training and also reveals the inefficiency of naive pixel reconstruction on natural images." (Page 12) What do you mean b y "inefficiency"? Data inefficiency? Computational inefficiency?

Without proper scientific language, none of these vague ideas can be quantified or tested. They simply amount to intuitions which may or may not be correct.

---

> ### Author Response · Authors · 2022-10-24
> **Author Response to Reviewer zo7M [1/4]**
>
> Thanks for the detailed and high-quality feedback. We really appreciate your recognition of the results, as well as your concerns on the motivation and using CLIP features. Our responses to your concerns are provided below.
>
> Since there exists a misunderstanding of our method and the use of CLIP features, we first answer your concerns and demonstrate that CrossMAE easily out-performs existing MIM regimes without extra knowledge or benefits (Q1-Q3). We also provide more context on the important of feature prediction and validate our motivation in Q4-Q8. We further study the importance of noise modeling in Q9-Q10.
>
> ##### **Q1: A possible unfair comparison using CLIP features, since other baselines are not using extra knowledge.**
>
> **A1:** We would like to emphasize that using CLIP is actually not a key part of our contribution. CrossMAE could be understood as a special way to learn from data with versatile targets. In fact, we’ve already provided the strong results with different prediction targets in Table 1, including pixels, EMA features and CLIP. You can see that without using extra knowledge, our method already out-performed these competitors by a clear margin.
>
> | Method           | PT Epochs               | FT (\%)              | Lin (\%)       |
> | --------------- | :------------------: |:------------------: |:------------------: |
> | MAE     |     300           |       83.1         | 	61.5	|
> | MAE     |     1600           |       83.6         | 	67.8	|
> | CrossMAE w Pixel |    **300**     |    **83.8**       |   **70.9**     |
>
> By putting the results in the same table, you can see that even with raw pixels, CrossMAE outperforms MAE significantly and can be proved as a more powerful and efficient pre-training scheme than previous methods. We believe this improvement is non-trivial. While we’ve already included this context in the paper, we agree with you that the current representation might hinder the contribution and are updating the draft to make it more clear.
>
> ##### **Q2: In Table 9, the use of CLIP has a strong impact on the performance.**
>
> **A2.1:** As mentioned in Sec 4.4, we only pre-trained with these different targets in 100 epochs, which is much shorter than MaskFeat (3 $\times$, 8 $\times$), MAE (16 $\times$), etc. The reason is: unlike evaluating on other downstream tasks, **each line with different targets in Table 9 requires a pre-training and fine-tuning process.** If we use the original settings, **we need to spend $n \times$ more computation on pre-training with $n \times$ targets**. Therefore, we chose a relatively smaller scale to allow fast and fair evaluation within our computation budgets. In this budgeted setting, CLIP targets can definitely have advantages due to its good zero-shot and few-shot performance, especially on classification. (additional context is provided in Q6)
>
> **A2.2:** To further address your concerns, we further **scale the experiments consistently to 300 epoch in these comparisons.** The consistent improvements better validates the generalization of CrossMAE, as well as the advantage of CrossMAE over other pre-training methods.
>
> | Method           | PT Epochs      | Targets           | Acc (\%)    |
> | :---------------: | :------------------: |:------------------: |:------------------: |
> | MAE     |     300           |     Pixel	|   83.1         |
> | CrossMAE     |     300           |     Pixel	|   **83.8**        |
> | MaskFeat     |     300           |      HOG         |   83.6	|
> |  CrossMAE   |     300           |       HOG        |  **84.2**	|
>
> ##### **Q3:Comparison with MAE given CLIP targets:**
>
> **A3:** Since the implementation of MaskFeat is not available, to better address your concern, we provide the performance of MAE with CLIP targets. All settings are kept consistent with former experiments. We can clearly find that CrossMAE serves as a powerful MIM paradigm, with better capability of learning from data.
>
> | Method           | Accuracy (\%)    |
> | :---------------: | :------------------: |
> | MAE + CLIP     |     84.1           |
> | CrossMAE    |     **84.8**       |

---

> > ### Author Response · Authors · 2022-10-24
> > **Author Response to Reviewer zo7M [2/4]**
> >
> > Although we’ve included the performance from feature prediction without extra knowledge in Table 1, we agree it might be easily neglected and would be instead focused on CLIP’s performance. Here we provide more details.
> >
> > ##### **Q4:Feature prediction without extra knowledge:**
> >
> > **A4:** While we believe the experiments provided have demonstrated CrossMAE’s capability without extra knowledge, *feature alignment is not leveraged when predicting low-level descriptors such as raw pixel and HOG*. Therefore, we’ve design **an extra settings that induce better metric space without extra knowledge such as CLIP:**. (Note that this is provided in the original submission in Table 1)
> >
> >  -  EMA features: we encode the target unmasked representation using an *exponentially moving average* (EMA) of the transformer backbone (online network), and let the CrossMAE system to predict target representation. This design follows the strategy in BYOL[1] and recent data2vec[2], which has been analyzed to improve feature fineness[3,4].
> >
> > | Method            | Targets    |Acc (\%)    |
> > | :---------------: |:---------------: |:-------------: |
> > | Baseline     |    Pixel	|   83.1         |
> > | CrossMAE    |     Pixel	|   **83.8**        |
> > | CrossMAE    |      EMA features         |   84.4	|
> > | CrossMAE  |       CLIP        |  **84.8**	|
> >
> > From the comparisons, we can further **understand how feature prediction without extra prior helps MIM pre-training.** The variant with an unsupervised target performs close to the CLIP target. These comparisons ensure fair comparison and indicate the benefit of feature alignment by cross prediction.
> >
> > **Q6: Downstream performance without extra knowledge**:
> >
> > **A6.1:** To further address your concerns on our motivations, we adopt this unsupervised feature as targets and ensure no additional data and labels are leveraged.  Here we adopt the CrossMAE with and without CLIP features and compare it to the MIM baseline. While CLIP brings improvements on classification on low-data regimes, you can see that without CLIP, CrossMAE clearly beats the baseline and performs close to CLIP. On other downstream tasks such as detection, the method even performs better. **The results demonstrate that our method is competitive without extra knowledge, given the benefits from feature alignment.**
> >
> > |                  |  ImageNet | ImageNet |  COCO    |	 COCO    | ADE20k    |
> > |-------------------------|:-----------------:|:------------:|:------------:|:------------:|:------------:|
> > |    **Metrics**        |   **PT 1\%**  | **PT 10\%**  | **AP box**    | **AP mask**    |  **mIoU**    |
> > |    Baseline      |    74.4       |       74.9       |      48.4       |   42.6       |    48.1     |
> > |    CrossMAE w CLIP    |      **77.6**      |      **78.1**       |  50.9        |   45.2        |      **50.4**         |
> > |    CrossMAE wo CLIP      |       77.1      |       77.7       | **51.1**        |   **45.3**        |      50.2         |
> >
> >
> > **Q7: Improvement on downstream tasks are among the similar magnitude of image classification**
> >
> > **A7:** Firstly, we believe the consistent improvement is non-trivial across a wide variety of tasks, which suggest the generalization of CrossMAE. Note that there exists many hyper-parameters in downstream tasks comparing to classification, while we keep them consistent with the baselines.As mentioned in Sec 4.4,  we’ve also mentioned that better downstream results are expected with a suitable target: : for example, on facial landmark detection, VGG features out-performed CLIP by the NME(&darr;) of **4.07** (CLIP: 4.21). On fine-grained classification, HOG worked slightly better. On object detection and instance segmentation, EMA encoder serves as a competitive scheme.
> >
> > Since we didn’t want the feature target to be a heuristic choice, which might be mis-understood as an important hyper-parameter by the readers. Also, we were unable to extensively evaluate downstream performance with all different targets. Therefore, we choose a relatively consistent target such as CLIP to report in most experiments, which could better convey the general idea of CrossMAE: disentangling low-level generation and high-level understanding. Meanwhile, choosing the most suitable target for different downstream tasks is actually out of the context of this paper. As potential relations exist in various targets and downstream tasks, we would like to leave this as an interesting takeaway and put it in our future work.

---

> > > ### Author Response · Authors · 2022-10-24
> > > **Author Response to Reviewer zo7M [3/4]**
> > >
> > > **Q8: Investigation of the learned features**
> > >
> > > **A8.1:** While with the additional discussion above, we believe it’s relatively more clear to show that motivation of our work and benefits from feature prediction. However, actual investigation should be helpful in understaning our method. One important quantitative metric to test the linear seperability of features has already been reported in Table 1, i.e. the linear probing accuracy. You can see that while major MIM-based methods perform poorly, CrossMAE obtain much better linear seperability.
> > >
> > > **A8.2:** To better investigate the features qualitatively, we randomly select 5000 images of 20 classes from the ImageNet validation set. Then we visualize the t-SNE feature after pre-training in this [Anonymous Link](https://anonymous.4open.science/r/CrossMAE_visualization-2676/README.md). Each point denotes one sample, of which the color denotes its ground truth class. On the left, we provide MAE as a comparison. You can clearly see that after pre-training, the feature distributions obtain better alignment and distinction between different semantics. The distance between different clusters is enlarged.
> > >
> > > ##### **Q9: Justify the important of noise modeling**
> > >
> > > **A9:** We agree with you that it’s important to provide more justification on this design. In Table 8, we would like to mention that, in the **+ Noise** columns, *Concatenation* doesn’t benefit, which suggests that directly concat a simple noise token to the inputs brings no improvement. We empirically find that this operation contributes subtle diversity in drawing head, similarly as you’ve concerned. Therefore we propose our noise mapping network that clearly introduces randomness and diversity (more analysis in Q10).
> > >
> > > Firstly, we believe that 0.5% elevation on ImageNet fine-tuning is solid and demonstrates the importance of this module, considering its simple design. Meanwhile, we totally agree with you that only with this accuracy boost, it might not be concrete enough to study the exact improvements with diversity on representation learning. With millions of labeled images provided in ImageNet fine-tuning, it would be difficult to study the generalization. Therefore, we provide two additional experiments to help better understand noise modeling. Firstly, we **compare the performance with limited percentage of labeled data in fine-tuning**, following the settings in Table 7:
> > >
> > > | Model           | 1%                | 10%               |
> > > | :---------------: | :------------------: |:------------------: |
> > > | wo Noise Modeling     |     62.1           |       75.3         |
> > > | **Ours**|    **64.1** (+2.0)       |   **76.5** (+1.2)     |
> > >
> > > From this table, we can see that noise modeling consistently boosts the baseline under limited labeled data setting. When the number of labeled images is smaller, the improvement is even larger. The results demonstrate that with noise modeling, the pre-trained model has **better generalization and transfer capability.**
> > >
> > > | Model           | mCE (&darr;)        |
> > > | :---------------: | :------------------: |
> > > | Baseline     |     51.7           |
> > > | + **Noise Modeling** |    **48.9**       |
> > >
> > > We further testify the performance of pretrain-fineuned ImageNet-1k model on ImageNet-C[5], to validate whether the noise modeling enhances the model’s **invariance and robustness towards texture change**. We believe the efficiency and robustness evaluation demonstrate the benefit of noise modeling in learning a better representation, from two perspectives.

---

> > > > ### Author Response · Authors · 2022-10-24
> > > > **Author Response to Reviewer zo7M [4/4]**
> > > >
> > > > #### **Q10: How does $E_{draw}$ models diverse possible completions as it was trained to predict $\hat{x}$?**
> > > >
> > > > **A10:** Motivated by recent studies of GANs[6,7] on generating diversity, noise modeling is one of the core designs of CrossMAE. Instead of modeling $p(\hat{x}|x_M)$, noise injection enables the drawing head to model the conditional probability of $p(\hat{x}|x_M, z)$. In other words, it maps noise distribution to a high-dimensional complex space. We randomly sample different $z$ during training and observe stochastic variations produced by the head.
> > > >
> > > > A nature concern is that the head might simply neglect the noise and still model $p(\hat{x}|x_M)$, which has been observed in generative modeling. Therefore we design the noise mapping network, to ensure randomness in the process[6]. We quantitative showed the benefits of this design in ablations (Table 8) and also provide the difference between sampled completions in Figure 3. As you’ve questioned, there exists no constraints or loss function to pay attention to the noise. However, this technique has been widely adopted in Style-based generator [7] and many applications of generative modeling, where the core idea is to map the noise through deeper and multi-scale stochastic variations. Our results also exhibit the huge difference. Recently, there have been some works studying this curious effects[6] in generative modeling, which reveals that noise injection as a kind of reparameterization in Euclidean spaces.
> > > >
> > > >
> > > > Thanks again for your detailed comments and thoughtful concerns. Apart from the additional discussion and studies we provided, we are also updating our draft based on your comments including the unscientific language throughout the paper, inaccurate statements and typos. If there exists any additional questions, we would also like to follow up.
> > > >
> > > >
> > > >
> > > > [1] BYOL: Bootstrap Your Own Latent: A New Approach to Self-Supervised Learning. NeurIPS 2020.
> > > >
> > > > [2] data2vec: A General Framework for Self-supervised Learning in Speech, Vision and Language. ICML 2022
> > > >
> > > > [3] Understanding Dimensional Collapse in Contrastive Self-supervised Learning. ICML 2021
> > > >
> > > > [4] Exploring Simple Siamese Representation Learning. CVPR 2021
> > > >
> > > > [5] Benchmarking Neural Network Robustness to Common Corruptions and Perturbations. ICLR 2019
> > > >
> > > > [6] Understanding Noise Injection in GANs. ICML 2021
> > > >
> > > > [7] ​​A Style-Based Generator Architecture for Generative Adversarial Networks. CVPR 2019

---

> > > > ### Comment · Reviewer_zo7M · 2022-10-28
> > > > **Thank you for your response**
> > > >
> > > > I appreciate the additional ablation studies.
> > > >
> > > > My main concern with the paper still stands. While CrossMAE does appear to get better performance, I am still unconvinced by the author's justifications for its improved performance. As it stands, the authors have demonstrated that each component is useful. However, I believe that some of the components still warrant further investigation.
> > > >
> > > > > Q7: Improvement on downstream tasks are among the similar magnitude of image classification
> > > >
> > > > I'm afraid you didn't answer my question here. You motivate CrossMAE as a model that will be better equipped at fine-tuned tasks. However, from the results, it seems like any improvements on fined-tuned tasks are highly correlated with the improvements on standard image classification. This begs the question: can CrossMAEs' improvements on fine-tuned tasks be completely explained by the fact that it is a "better" model (as measured by its performance on image classification tasks)? If so, the justification for the CrossMAE loss (that it is somehow better suited for learning fine-tunable representations) is flawed.
> > > >
> > > > > As you’ve questioned, there exists no constraints or loss function to pay attention to the noise. However, this technique has been widely adopted in Style-based generator [7] and many applications of generative modeling, where the core idea is to map the noise through deeper and multi-scale stochastic variations.
> > > >
> > > > If nothing is constraining the model to pay attention to the noise, why does including noise benefit the model? I am completely unconvinced that this is because the noise is injecting "diversity." Could it be simply adding more stochasticity to the training process, allowing the model to converge to a "flatter" minimum (https://arxiv.org/abs/1609.04836)? Would results improve if there was actually a constraint in the loss function that required the network to incorporate the noise?
> > > >
> > > > -------
> > > >
> > > > Overall, think that the CrossMAE model is promising, and it is worth sharing these improvements with the community. At the same time, the author's justifications/explanations for improved performance lack scientific rigor, which is a pre-requisite for publication in these proceedings. I would encourage the authors to revise the paper; focusing on understanding *why* the various mechanisms improve performance, rather than simply running ablation studies and reporting SOTA results.

---

> > > > > ### Author Response · Authors · 2022-10-29
> > > > > **Thanks for your response**
> > > > >
> > > > > Thanks a lot for your feedback! We would also like to follow up with some of your concerns
> > > > > ```
> > > > > Q7: Improvement on downstream tasks are among the similar magnitude of image classification
> > > > > ```
> > > > > Sorry for the confusion here.
> > > > >
> > > > >  - As we misunderstood your questions before, we would like to mention that the common definition of downstream tasks for self-supervised methods includes image classification (fine-tuning, as well as linear probing), segmentation and detection, etc. Note that image classification is precisely treated as one important downstream task in existing self-supervised learning papers. We mentioned and showed that our pre-training method has strong and generalizable performance across these tasks.
> > > > >
> > > > >  - Every task in our experiment follows first: **CrossMAE Pre-training**, then: **Downstream evaluation**, where the intermediate fine-tuning on classification is not involved. It's also non-trivial to obtain consistent improvement across different tasks or compare the magnitude here, e.g. *MAE is evaluated on classification downstream tasks only, BEiT is evaluated on fine-tuning and segmentation only, with relatively lowered performance on detection*) As you've mentioned, we also believe that the improvements in these tasks validate our pre-training method leads to a good and transferable representation.
> > > > >
> > > > > ```
> > > > > If nothing is constraining the model to pay attention to the noise, why does including noise benefit the model
> > > > > ```
> > > > >
> > > > >  - Thanks for pointing it out. We agree with you that there require theories and concrete investigations to support the benefit of noise injection and are also currently working on updating with further details. This multi-level noise injection has been widely adopted in SOTA generative model variants, without theoretical analysis and investigation, e.g. BigGAN splits input vectors into one chunk per layer and projects each chunk to the gains and biases of batch normalization in each layer, where they claim such design allows direct influence on features at different resolutions and levels of hierarchy. StyleGAN, on the other hand, introduces noise injection for multi-scale stochastic variations.
> > > > >
> > > > >  - Our design choice originally targets providing more stochastic diversity to the drawing outputs, therefore adopting the multi-level noise injection. Due to the empirical success and wide-adoption of these designs, we initially thought it was a little bit out of the context of our work. A recent work [1] analytically shows that the expressive power of generator is possibly limited by the rank of its jacobian matrix, which decreases monotonically and could be alleviated by noise injection. We would like to incorporate more analysis for the noise modeling design, which would be definitely helpful in understanding this merely-understood mechanism.
> > > > >
> > > > > Thanks again for your valuable comments!
> > > > >
> > > > > [1] Understanding Noise Injection in GANs. ICML 2021

---

> > > > > ### Author Response · Authors · 2022-10-31
> > > > > **Thanks for your response and Further inverstigation on the noise injection**
> > > > >
> > > > > Thanks again for your helpful feedback. Based on your suggestion about noise injection, we have also conducted further studies on the effect of noise injection in our network.
> > > > >
> > > > > ```
> > > > > Could it be simply adding more stochasticity to the training process, allowing the model to converge to a "flatter" minimum?
> > > > > ```
> > > > > In response to this, we evaluated the activation of the drawing head with and without the noise mapping network:
> > > > >  - when simply attaching the noise vector to the input, the network tends to simply neglect the noise, i.e. if we change the sampled noise, the activation and network outputs basically remain unperturbed.
> > > > >  - when applying transformations to the noise using the noise mapping network into multi-scale noise tokens, given the same masked images, different sampled noise leads to completely different network activation, as well as different output as shown in Figure. 3.
> > > > >
> > > > > These observations show that such noise injection could be understood as re-parameterization on the network. In our case, given the lightweight architecture of the drawing head, it leads to stochasticity in training and sampling.
> > > > >
> > > > > ```
> > > > > Would results improve if there was actually a constraint in the loss function that required the network to incorporate the noise?
> > > > > ```
> > > > > In response to your suggestion, we adopt the mode-seeking regularization term in MS-GAN [1] as an additional loss that requires output to be more associated with the noise context.
> > > > >
> > > > > Given two sampled noises and corresponding output images, this term proposes to maximize the ratio of the distance between output images with respect to the distance between noise vectors. Intuitively, it tries to push to model to generate dissimilar images given different noises.
> > > > >
> > > > > With this loss, we find that the sampled diversity is further improved. Since the drawing head doesn't focus on generating more realistic samples but on better diversity, we find that this improved diversity in this stage improves the training of the backbone, by measuring the model’s invariance and robustness towards texture change on ImageNet-C.
> > > > >
> > > > > | Model           | mCE (&darr;)        |
> > > > > | :---------------: | :------------------: |
> > > > > | Baseline     |     51.7           |
> > > > > | + **Noise Modeling** |    **48.9**       |
> > > > > | + **Loss** |    **48.1**       |
> > > > >
> > > > > The above evaluations show that the noise mapping network could be understood as one re-parameterization on the drawing head, which provides more stochasticity. With a loss function, the modeling diversity from the conditional distribution is better captured. We hope this discussion better explains the role of our noise injection design.
> > > > >
> > > > > Thanks again for your suggestion!
> > > > >
> > > > > [1] Mode Seeking Generative Adversarial Networks for Diverse Image Synthesis. CVPR 2019

---

### Review · Reviewer_oZYh · 2022-10-09

**Summary Of Contributions:**

The paper proposes a framework for supervised learning with image data, called CrossMAE. For a given supervised learning task, the CrossMAE framework entails two steps. The first step, called mask-then-draw, performs a small perturbation of each image in the dataset. The second step, called draw-then-predict, learns a classifier/regressor from a transformer architecture, using the perturbed images as input and the original targets of supervised learnings as outputs. Experiments are done on a variety of supervised learning tasks (image classification, object detection, instance segmentation etc). The numerical results show that CrossMAE yields accurate predictions, often outperforming existing techniques.

**Requested Changes:**

Here are some changes I would like to see in the next draft.

- Write out the meaning of the MAE acronym.
- Justify the need for learning the drawing head, since there are cheaper alternatives available.
- Discuss experimental details in a self-contained manner
- Explain why Table 10 does not contain runtime comparisons between CrossMAE and the competitor techniques from previous tables. If there is not a strong reason for the omission, I would like to see inclusion of the runtime comparisons.

**Strengths And Weaknesses:**

# Strength
The steps involved in CrossMAE are well-explained: the accompanying figures (Figure 1 and Figure 2) are very helpful, and the amount of technical detail/equations is just right.

The authors produce a plethora of empirical results (Tables 1 through Table 5) to support the claim that CrossMAE improves over existing masked image modelling (MIM) approaches in terms of accuracy metrics.

# Weaknesses

The paper does not explain how hyperparameters are picked. The set of hyperparameters include $\lambda$ of Equation 4 or $\sigma$ in the Gaussian filter that creates the coarse reconstruction. I did not see discussions of these hyperapameters in, say, the ``implementation details'' paragraph of Section 4.1.

The paper is not self-contained in key places. For instance, the ``implementation details'' paragraph refers to external references when discussing experimental setup. Another instance is the use of the acronym MAE without explanation.

It is not discussed why the drawing head (Section 3.1) is necessary, in the sense that, one could seemingly just use the smoothed target $\hat{x}$ instead of $x_{draw}$ as inputs of the backbone $f_{net}$. Just using $x_{draw}$ would save the compute time necessary to train the drawing head.

Table 10 does not contain the runtime comparison between CrossMAE and the competitor techniques from previous tables (say Table 1 through 5). I am willing to believe that there is a good reason for omitting the competitors, but I would like to see it spelled out.

---

> ### Author Response · Authors · 2022-10-24
> **Author Response Reviewer oZYh [1/2]**
>
> Thanks for your detailed feedback. We really appreciate your recognition of our explanation and experiments, as well as your concerns on some pivotal points. Our responses to your concerns and requested changes are provided below.
>
> ##### **Q1: Write out the meaning of the MAE acronym**
>
> **A1:** Thanks for pointing it out. We have updated the our draft about the acronyms used including BERT[1],  MAE[2], etc, to make our paper more self-contained.
>
> ##### **Q2: Justify the need for learning the drawing head**
>
> **A2:** Since the drawing head and noise modeling are the key components in our design, we agree with you that we need more elaborations on this module.
>
> **A2.1: The advantage of drawing head:** The drawing head design in our work helps representation learning in two aspects:
> 1. With noise mapping network, it models one-to-many mapping and can mitigate problems of hard memorization, which we expect the model leads to better diversity and generalization.
> 2. As shown in Figure. 3(d), the drawing head provides diverse completions with varying textures. Therefore, these sampled coarse outputs also strengthen the backbone with better visual invariance towards texture.
>
> The noise modeling design improves the feature learning and generalization of the transformer backbone, as well as its texture invariance and robustness. We provided the ablations in Table 8 with the baseline+drawing (83.3) versus baseline+drawing+noise (**83.8**), from which you can see a clear boost from this simple design.  We also find that simply proposing the drawing head without noise, the results are not improved. Meanwhile, we totally agree that only with this accuracy boost, it might not be concrete enough to study the exact improvements with diversity on representation learning. With millions of labeled images provided in ImageNet fine-tuning, it would be difficult to study the generalization.
>
>  The experiment settings follow Section 4.1 and Table 7, where we also show the results using limited labeled fine-tuning data (1%, 10%) in ImageNet. The noise modeling consistently helps performance and improves the accuracy more under limited data. These results can further demonstrate the boosted generalization.
>
> | Percentage           | wo Noise                |  **Ours**   |	 Delta	|
> | --------------- | :------------------: |:------------------: |:------------------: |
> | 1%     |     62.1           |      **64.1**      | +2.0    |
> | 10% |    75.3       |   **76.5**    |	+1.2 	|
>
> As shown in Figure. 3(d), the drawing head provides diverse completions with varying textures.  We further testify the performance of pretrain-fineuned ImageNet-1k model on ImageNet-C [3], to validate whether the noise modeling enhances the model’s invariance and robustness towards texture change.
>
> | Model           | mCE (&darr;)        |
> | :---------------: | :------------------: |
> | Baseline     |     51.7           |
> | + **Noise Modeling** |    **48.9**       |
>
> **A2.2: The need for learning the drawing head:**  We would like to recap that the light-weight drawing head incurs limited computation overheads to the system, as we do not expect this module to perform nice in-painting given smoothed targets. As demonstrated in Figure 3, the sampled mask-then-draw process produces diverse blur and distorted completions. The coarse target is still difficult and only serves as a proxy target for our drawing head. We can also understand the drawing head as a module that performs sophisticated and diverse online augmentation on masked regions. As suggested, we provide the results by replacing the inputs with Gaussian-smoothed targets. You can see that simply smoothing the inputs has no influence on the results, while our learnable drawing head benefits a lot. Comparing to simply blurring the input images, the drawing head provides more complex modeling and variance.
>
> | Method           | Accuracy      |
> |---------------| :------------------: |
> | Original Image   |     84.0           |
> | Gaussian Smoothed    |     83.9           |
> | Ours (drawing head)     |     **84.7**           |

---

> > ### Author Response · Authors · 2022-10-24
> > **Author Response Reviewer oZYh [2/2]**
> >
> > ##### **Q3: Discuss experimental details in a self-contained manner.**
> >
> > **A3:** Thanks for mentioning this, we discuss the experimental setup and hyper-parameters in the Appendix.
> >
> > ##### **Q4: Explain why Table 10 does not contain comparisons towards other competitors.**
> >
> > **A4.1:** Our system uses most experimental settings and hyper-parameters as in MAE, and adopt MAE as our major baseline. Since CrossMAE proposes a mask-then-draw step that introduces additional costs, we think it’s necessary to directly compare with MAE in terms of efficiancy. Therefore, we report the runtime comparison against MAE to show our advantage.
> >
> > **A4.2:** Meanwhile, it’s definitely meaningful to compare with other competitors in Table 1. However, some of these methods adopted different schedules and haven’t open-sourced their implementations yet. In addition, MAE have already shown better training efficiency than other BEiT-based MIM methods and contrastive methods, as mentioned in the paper[2]. Therefore, we choose to directly compare with MAE in our paper. However, we would also like to compare with other methods that have released their implementations if needed. Following the settings in Section 4.5, we also re-implement BEiT and CAE training with the same hardware, where you can clearly see the benefits of our method.
> >
> > | Method           | PT Epochs      | Wall-clock Time      |
> > |---------------| :------------------: | :------------------: |
> > | BEiT   |    800         |      246h           |
> > | CAE    |     800           |      280h           |
> > | MAE    |     1600           |     464h           |
> > | CrossMAE    |     300          |      107h           |
> >
> > ##### **Q5: Elaborate on how are the hyper-parameters including $\lambda$ and $\sigma$ chosen.**
> >
> > **A5:** As mentioned in Page 17 of **Balancing Loss weights**, we originally set the hyper-parameters based on balancing these two losses. We’ve found that as far as we choose relatively reasonable hyper-parameters for both $\lambda$ and $\sigma$, the whole system works fine. Since the difficulty of draw-then-predict occasionally leads to higher losses, we set $\lambda$ to 0.1 and $\sigma$ to 10. To provide you with additional context, we sweep different loss weights and kernels.
> >
> > | $\lambda$ | Accuracy      |
> > |---------------| :------------------: |
> > | 0.1   |    84.6         |
> > | 0.4    |     84.7           |
> > | 1.0    |     84.5           |
> > | 10    |     diverge          |
> >
> > Due to limited computation resources, we are unable to spend much effort on sweeping a perfect set of hyper-parameters. However, You can find that the results are relatively consistent given proper hyper-parameters.
> >
> > | $\sigma$ | Accuracy      |
> > |---------------| :------------------: |
> > | 1   |   84.2        |
> > | 5    |     84.6           |
> > | 10    |     84.7           |
> > |   20  |     84.8          |
> >
> >
> > Thanks again for your comments on our paper. We are updating our draft by including the additional requested changes based on your valuable comments, including the implementation details and the justification of the drawing head. If there exists any additional questions, we would also like to follow up.
> >
> > [1] BERT: Pre-training of Deep Bidirectional Transformers for Language Understanding. NAACL 2019
> >
> > [2] Masked Autoencoders Are Scalable Vision Learners. CVPR 2022
> >
> > [3] Benchmarking Neural Network Robustness to Common Corruptions and Perturbations. ICLR 2019

---

### Review · Reviewer_qt55 · 2022-10-13

**Summary Of Contributions:**

Large-scale self-supervised pretraining has greatly benefited image modeling and obtained impressive performance on image-related downstream tasks. Still, how to design effective and efficient pretraining objectives is a big research question. In this paper, the authors propose to decompose the widely used “mask-then-predict” paradigm into two separate parts: “mask-then-draw” and then “draw-then-predict”. The former is designed to be lightweight, perform low-level generation without reconstructing all image details, and produce coarse images with a certain degree of diversity; those coarse images are transferred to the latter to perform high-level semantic understanding and predict the targets. This new paradigm was believed to disentangle the generation and understanding responsibility and mitigates the discrepancy between pretraining and finetuning, which would improve the expressiveness of the learned image representations. Extensive experiments on several tasks show that the proposed method, CrossMAE, obtained better performance than existing methods; further ablation study also supports this new paradigm.


**Broader Impact Concerns:**

I didn't find big problems.

**Requested Changes:**

While the performance is impressive, I find that where the improvements actually come from requires further investigation. My major concerns are given below:

1) When using CLIP as the target, CrossMAE implicitly becomes a student model which learns the knowledge from CLIP. In particular, CLIP was reported to achieve a zero-shot top-1 accuracy of 76.2 on ImageNet. Then what if using CLIP directly for the downstream task (finetuning and linear probing)? What’s the advantage of CrossMAE over CLIP?

2) Also, results in Table 1 show that the main improvement on Fine-tune comes from the use of CLIP.

3) Based on the analysis in Table 8, we can find that adding noise doesn’t benefit the performance that much. This conflicts with the initial assumption that the drawing head delivers diverse and ambiguous outputs that enforces visual invariance to the backbone so benefit the pretraining. Do we really need diversity here? Table 8 shows that the most benefits are from the use of coarse image and the predictor, rather than the diversity. What if dropping the drawing head and replacing the masked image with the Gaussian-filtered coarse image (perhaps adding some noise during the filtering to add diversity)? This makes sense, particularly when using CLIP as the target: the backbone tries to predict semantic features based on the noised image.

4) Further, the authors argue that CrossMAE disentangles its generation and understanding. This is misleading since the “draw-then-predict” step still performs some sort of generation, for example when the pixel is used as the target.


**Strengths And Weaknesses:**

Strengths:

The authors propose a simple and effective pertaining method for image modeling, called CrossMAE, that decomposes "mask-then-predict" into two steps "mask-then-draw" and "draw-then-predict". CrossMAE obtains improved performance across several downstream tasks compared to several strong baselines.

Weaknesses:

It's still not clear to me whether the decomposition is helpful since there are many other design choices in CrossMAE that might explain the quality improvements.

---

> ### Author Response · Authors · 2022-10-24
> **Author Response to Reviewer qt55 [1/2]**
>
> We appreciate the constructive comments provided and sincerely thank you for comprehensive understanding of this work from multiple aspects. Our response to your concerns is given below:
> ##### **Q1: When using CLIP as the target, CrossMAE becomes a student learning from CLIP. What if we directly use CLIP for fine-tuning and linear probing? And what’s the advantage of CrossMAE over CLIP.**
>
> **A1:** Thanks for raising this interesting point. Yes, we agree with you that when using CLIP, CrossMAE can be understood as a special way to learn from CLIP models. As CrossMAE serves as a pre-training system from versatile targets, including pixels or features, we have viewed it as a special and better way of exploiting them.
>  - **Comparison with directly using CLIP**: To better understand the contribution of our method, we also provide the direct fine-tuning and linear probing of CLIP’s ViT-B/16 vision encoder, with the exact same settings in Table 1. We can see that with CrossMAE pre-training, the model has a clear higher fine-tuning and lowered linear probing accuracies. As CLIP is trained with large-scale language-text pairs through contrastive learning, it shows better linear and zero-shot performance compared to MIM-based methods as expected. However, our CrossMAE pre-training leverages the CLIP target significantly better and learns representations that maintain the finetuning-linear trade-off, which also suggests the potential of masked modeling.
>
> | Method           | Fine-tune      | 	Linear        |
> |---------------| :------------------: |:------------------: |
> | MAE     |     83.6           |     67.8	|
> | CLIP     |     82.9           |     **79.5**	|
> | w CrossMAE     |     **84.7**           |     76.3	|
>
>  - **Advantage over CLIP:** More importantly, the major takeaway of our work is not to choose the very best prediction target, such as CLIP features, but to provide a novel MIM pre-training paradigm where low-level generation and high-level understanding are separated. The CLIP vision encoder here serves as an example of high-level extractor with high-level semantics, while we also experiment with different prediction targets.  As we’ve demonstrated that pixels, HOG, or features from momentum encoder benefit from our CrossMAE paradigm. While our focus is to motivate proper MIM design, we believe such advances in finding proper targets beyond pixels can further benefit CrossMAE.
>
> ##### **Q2: From Table 8, adding noise doesn’t seem to benefit the performance that much.**
> **A2:** Thanks for your important question. In Table 8, we would like to mention that, in the **+ Noise** columns, *Concatenation* doesn’t benefit, which suggests that directly adding noise token brings no improvement. We empirically find that the naive single-level noise concatenation contributes subtle diversity in drawing head, therefore proposing our noise mapping network that clearly introduces randomness and diversity.
>
> Firstly, we believe that 0.5% elevation on ImageNet fine-tuning is solid and demonstrates the importance of this module, considering its simple design. Meanwhile, we totally agree with you that only with this accuracy boost, it might not be concrete enough to study the exact improvements with diversity on representation learning. With millions of labeled images provided in ImageNet fine-tuning, it would be difficult to study the generalization. Therefore, we provide two additional experiments to help better understand noise modeling. Firstly, we **compare the performance with limited percentage of labeled data in fine-tuning**, following the settings in Table 7:
>
> | Model           | 1%                | 10%               |
> | :---------------: | :------------------: |:------------------: |
> | wo Noise Modeling     |     62.1           |       75.3         |
> | **Ours**|    **64.1** (+2.0)       |   **76.5** (+1.2)     |
>
> From this table, we can see that noise modeling consistently boosts the baseline under limited labeled data setting. When the number of labeled images is smaller, the improvement is even larger. The results demonstrate that with noise modeling, the pre-trained model has **better generalization and transfer capability.**
>
> | Model           | mCE (&darr;)        |
> | :---------------: | :------------------: |
> | Baseline     |     51.7           |
> | + **Noise Modeling** |    **48.9**       |
>
> We further testify the performance of pretrain-fineuned ImageNet-1k model on ImageNet-C[1], to validate whether the noise modeling enhances the model’s **invariance and robustness towards texture change**. We believe the efficiency and robustness evaluation demonstrate the benefit of noise modeling in learning a better representation, from two perspectives.

---

> > ### Author Response · Authors · 2022-10-24
> > **Author Response to Reviewer qt55 [2/2]**
> >
> > ##### **Q3: What if dropping the drawing head and replacing the masked image with the Gaussian-filtered coarse image?**
> > **A3:** The drawing head in our design is light-weight and incurs limited computation overheads to the system, as we do not expect this module to perform nice in-painting given smoothed targets. As demonstrated in Figure 3, the sampled mask-then-draw process produces diverse blur and distorted completions. The smoothed target is still difficult and only serves as a proxy target for our drawing head. We can also understand the drawing head as a module that performs sophisticated and diverse online augmentation on masked regions. As suggested, we provide the results by replacing the inputs with Gaussian-smoothed targets. You can see that simply smoothing the inputs has no influence on the results, while our learnable drawing head benefits a lot. Comparing to simply blurring the input images, the drawing head provides more complex modeling and variance.
> >
> > | Method           | Accuracy      |
> > |---------------| :------------------: |
> > | Original Image   |     84.0           |
> > | Gaussian Smoothed    |     83.9           |
> > | Ours (drawing head)     |     **84.7**           |
> >
> > ##### **Q4: The term of ``disentangle’’ is misleading.**
> > **A4:** Thanks for pointing it out. We agree with you that the term **disentangled** is a little misleading and unscientific, as our method does not explicitly separate low-level generation and high-level understanding. We would like to rephrase it as **decoupled** to better describe our method.
> >
> > Thanks so much for your detailed question. We are updating our draft by including the additional requested changes based on you valuable comments. If there exists any additional questions, we would also like to follow up.
> >
> > [1] Benchmarking Neural Network Robustness to Common Corruptions and Perturbations. ICLR 2019

---

> > > ### Comment · Reviewer_qt55 · 2022-11-02
> > > **Thanks for the updated results**
> > >
> > > Thanks for your response and the updated results. I still have some concerns, particularly regarding the necessity of the drawing head and the use of CLIP features as the target.
> > >
> > > Firstly, from the results in A1, it seems that the improvement in Linear Probing can be mostly explained by the use of CLIP. Could you please show the result for "MAE + CLIP" which would more directly reflect how CrossMAE works compared to MAE.
> > >
> > > Secondly, in Table 8, adding noise helps both concatenation and mapping network marginally. Whether a 0.5% improvement is solid should depend on how significant it is. I would suggest you add a significance test here. In addition, it's interesting to see that adding noise benefits data efficiency. Similarly, could you also perform this experiment for "CrossMAE + Pixel"? I want to know how much this data efficiency comes from CLIP.
> > >
> > > Thirdly, it's great to see using Gaussian smoothed input doesn't deliver similar benefits as using the drawing head. Apart from Accuracy, could you please also show the Linear Probing results? Also, the current description for "Gaussian Smoothed" is very vague. Please elaborate. Did you apply noise and prediction to "Gaussian Smoothed"? What's the target used? I want to make sure that the drawing head is indeed required as the visualization examples seem to be trivial to simulate.

---

> > > > ### Author Response · Authors · 2022-11-05
> > > > **Author Response to Reviewer qt55**
> > > >
> > > > Thanks for your response on the updated results and some useful concerns:
> > > >
> > > > ##### **Q: The results for MAE + CLIP to reflect how CrossMAE works compared to MAE**
> > > > **A:** Thanks for this important suggestion, we also reported the FT performance of MAE+CLIP in A3 to Reviewer zo7M. The linear and FT performance is also listed below:
> > > >
> > > > | Method           | Target   	|   PT Epoch       | FT (\%)    | Linear (\%)    |
> > > > | :---------------: | :------------------: | :------------------: | :------------------: | :------------------: |
> > > > | MAE    |     Pixel         |     300          |     83.2           | 	61.5	 |
> > > > | **CrossMAE**    |    Pixel      |     300         |     **83.8**           |   **70.9**          |
> > > > | MAE    |     CLIP         |    	300	| 84.1           |      72.2        |
> > > > | **CrossMAE**    |      CLIP         |     300	| **84.8**       |    **76.3**       |
> > > >
> > > > You can see that the improvement of CrossMAE on linear results is clear, for both pixel and CLIP targets. This comparison illustrates the importance of our design.
> > > >
> > > > ##### **Q: The results on adding noise benefits data efficiency for CrossMAE w pixel, to demonstrate how much data efficiency comes from CLIP**
> > > >
> > > > **A:**  Thanks for this suggest. We include the comparison with and without noise modeling on CrossMAE + Pixel. We can also clearly see the improvement in low-data regime brought by noise modeling. Obviously, using CLIP as target will further improve both the baseline and our method. However, the noise modeling is designed to better learn from them.
> > > >
> > > > | Method           | Target   	|  1%      | 10%    	|
> > > > | :---------------: | :------------------: | :------------------: | :------------------: |
> > > > | wo Noise    |     Pixel         |     61.4          |     74.2           |
> > > > | **w Noise**    |    Pixel      |     **62.8** (+1.3)         |    **75.7**  (+1.5)         |
> > > >
> > > >
> > > > ##### **Q: Elaborate on Gaussian Smoothed and report the performance of linear probing**
> > > > **A:** Sorry for not clearly describe the practice in A3. As suggested by you and mentioned by Q3/A3, we want to validate if the drawing could simply be replaced by gaussian smoothed inputs. Therefore, we remove the drawing head in CrossMAE w CLIP and use gaussian smoothed image as the inputs (target here uses CLIP). The “original image” here is similar to a MAE with CLIP target. From the results in A3, you can clearly see the importance of drawing head. As you’ve suggested, we also provide the linear probing results here, which is a bit lowed than using original image.
> > > >
> > > > | Inputs           | Linear Acc      |
> > > > |---------------| :------------------: |
> > > > | Original Image   |        72.0        |
> > > > | Gaussian Smoothed    |     71.8           |
> > > > | Ours (drawing head)     |     **76.3**           |
> > > >
> > > > Thanks a lot for your detailed reviews. Hope these updated results solve your concerns. If there exists any additional questions, we would also like to follow up.

---

### Author Response · Authors · 2022-10-27
**General Response to reviewers and ACs on the updated draft**

Thank every reviewer and ACs for your time and insightful reviews from different perspectives. We have replied to each reviewer to address the questions and concerns individually, as well as revise our draft with updated content.  In this general response, we highlight some important points to address several remarks.

 - Firstly, we update the experimental comparisons in Table 1 with pixel and feature targets for both ViT and Swin Transformer, in order to validate the effectiveness of CrossMAE paradigm and the importance of feature prediction. Without leveraging extra knowledge, CrossMAE still clearly outperforms previous methods. We would also like to emphasize that our target is to explore a different MIM pre-training scheme, which is not as intensively studied as MAE. Therefore, there exist many points to explore within our decoupled pipelines, such as finding a valid pre-training target, as we discussed in Sec.4.4 and Table 9.


 - Secondly, since the drawing head as well as noise modeling play a key role in our framework, we agree it’s important to quantify their effect. However, we found that simply measuring fine-tuning or linear accuracy might not directly reflect their benefits. Therefore, we provide two additional experiments to help better testify noise modeling, including 1. the performance with a limited percentage of labeled fine-tuning data in Table 10, 2. and validate whether the noise modeling enhances the model’s invariance and robustness towards texture change on ImageNet-C in Table 11. We believe the efficiency and robustness evaluation demonstrate the benefit of noise modeling in learning a better representation, from two perspectives.


 - Thirdly, we also include the discussion about loss weights in A.1 and Table 15. To make our work more self-contained, we also report the detailed hyper-parameters in A.3.

 - Last but least, we’ve also polished our draft against some imprecise, unscientific terms and typos. We’ve also modified the term ``disentangled’’ into ``decoupled’’ to avoid some confusion.

As the end of discussion window is approaching, we would like to hear from you and polish our draft based on your precious comments. Thanks again for your time and efforts in reviewing.

---

### Decision · Action_Editors · 2022-11-21

**Recommendation:** Reject

**Comment:**

Overall, it is a good paper: clear writing, good organization, and extensive experiments. This merit is well-recognized by all of the reviewers. During the discussion, one of the reviewers would like to see some insightful analysis of why the proposed noise smoothing stage: mask-then-draw, helps better representation learning. However, the authors didn't provide any justifications.

In the end, two reviewers were leaning to reject the paper due to the abovementioned flaw. AE read the paper, reviews, and discussions, and agree with the concerns raised by the reviewers. AE encourages the authors to address the comments and re-submit the paper.

**Audience:**

The paper presents an interesting two-stage improvement based on the existing MAE framework. Any practitioner/researcher who is working on visual self-supervised learning would be interested in this paper.

**Claims And Evidence:**

This paper provides extensive experimental results that justify the proposed method is better than the conventional MAE. However, the key drawback, as raised by all the reviewers, is that it lacks a theoretical analysis of why the proposed two-stage method is better.